# Daytime atmospheric oxidation capacity in four Chinese megacities during the photochemically polluted season: A case study based on box model simulation

Zhaofeng Tan[1,2], Keding Lu[1,*], Meiqing Jiang[1], Rong Su[1], Hongli Wang[3], Shengrong Lou[3], Qingyan Fu[4], Chongzhi Zhai[5], Qinwen Tan[6], Dingli Yue[7], Duohong Chen[7], Zhanshan Wang[8], Shaodong Xie[1], Limin Zeng[1], and Yuanhang Zhang[1,9,10,*]

[1]State Key Joint Laboratory of Environmental Simulation and Pollution Control, College of Environmental Sciences and Engineering, Peking University, Beijing 100871, China

[2]Institute of Energy and Climate Research, IEK-8: Troposphere, Forschungszentrum Jülich GmbH, Jülich, Germany

[3]State Environmental Protection Key Laboratory of Formation and Prevention of the Urban Air Complex, Shanghai Academy of Environmental Sciences, Shanghai 200233, China

[4]Shanghai Environmental Monitoring Center, Shanghai 200235, China

[5]Ecological and Environmental Monitoring Center of Chongqing, Chongqing 401147, China

[6]Chengdu Academy of Environmental Sciences, Chengdu 610072, China

[7]State Environmental Protection Key Laboratory of Regional Air Quality Monitoring, Guangdong Environmental Monitoring Center, Guangzhou 510308, China

[8]Beijing Key Laboratory of Atmospheric Particulate Monitoring Technology, Beijing Municipal Environmental Monitoring Center, Beijing 100048, China

[9]Beijing Innovation Center for Engineering Sciences and Advanced Technology, Peking University, 100871, Beijing, China

[10]CAS Center for Excellence in Regional Atmospheric Environment, Chinese Academy of Sciences, Xiamen, China

*Correspondence to*: *k.lu@pku.edu.cn; yhzhang@pku.edu.cn*

**Abstract:** Atmospheric oxidation capacity is the basis for converting freshly emitted substances into secondary products, and is dominated by reactions involving hydroxyl radicals (OH) during daytime. In this study, we present in-situ measurements of $RO_x$ radical (hydroxy OH, hydroperoxy $HO_2$, and organic peroxy $RO_2$) precursors and products; the measurements are carried out in four Chinese megacities (Beijing, Shanghai, Guangzhou, and Chongqing) during photochemically polluted seasons. The atmospheric oxidation capacity is evaluated using an observation-based model and radical chemistry precursor measurements as input. The radical budget analysis illustrates the importance of HONO and HCHO photolysis, which account for ~50% of the total primary radical sources. The radical propagation is efficient due to abundant NO in urban environments. Hence, the production rate of secondary

pollutants, that is, ozone (and fine particle precursors [$H_2SO_4$, $HNO_3$, and extremely low volatility organic compounds (ELVOCs)] is rapid, resulting in secondary air pollution. The ozone budget demonstrates its high production in urban areas; also, its rapid transport to downwind areas results in rapid increase in local ozone concentrations. The $O_3$–$NO_x$–VOC (volatile organic compound)sensitivity tests show that ozone production is VOC-limited, and that alkenes and aromatics should be mitigated first for ozone pollution control in the four studied megacities. In contrast, $NO_x$ emission control (that is, a decrease in $NO_x$) leads to more severe ozone pollution. With respect to fine-particle pollution, the role of the $HNO_3$–$NO_3$ partitioning system is investigated using a thermal dynamic model (ISORROPIA2). Under high relative humidity (RH) and ammonia-rich conditions, nitric acid converts into nitrates. This study highlights the efficient radical chemistry that maintains the atmospheric oxidation capacity in Chinese megacities and results in secondary pollution characterized by ozone and fine particles.

**1. Introduction**

Air pollution is one of the major threats to human health in cities (Kan et al., 2012). The rapid economic development in eastern China has been accompanied by air-quality degradation in the last decades (Chan and Yao, 2008). More than 300 million people live in the North China Plain (NCP), Yangtze River Delta (YRD), and Pearl River Delta (PRD) regions in eastern China. Beijing, Shanghai, and Guangzhou are metropolitan cities in these regions, which suffer from severe air pollution. The Chengdu–Chongqing city group (population: 90 million), located in the Sichuan Basin (SCB) in Southwest China, represents a developing city cluster. Chongqing is the biggest city in Southwest China, which also suffers from severe air pollution. To improve the air quality, emission mitigations have been implemented since the 2000s. As a result, primary pollutant concentrations have declined. However, secondary pollution characterized by high concentrations of ozone and fine particles has become the major contributor to air pollution. The major components of fine particles are secondary components (Tie et al., 2013; Sun et al., 2004, 2006; Huang et al., 2014; Guo et al., 2014; Cheng et al., 2016; Zheng et al., 2005; He et al., 2001) such as sulfate, nitrate, and oxidized organic aerosol. This indicates the high oxidation capacity of Chinese pollution environments. Nationwide measurements showed that ozone is the only substance among six air quality indexes that has increased during the last five years (Li et al., 2018c; Lu X. et al., 2018). It is therefore difficult to control secondary pollution

given the nonlinear relation between primary and secondary pollutants. In contrast, the ozone pollution has been mitigated in the last decades by efficient precursor emission control in the U.S. (Parrish et al., 2017). Efficient ozone pollution control requires the knowledge of the effect of oxidation processes on the formation of secondary pollution because the atmospheric oxidation capacity (AOC) plays the key role in converting primary pollutants into secondary ones.

So far, only few studies elucidating oxidation processes were performed in China (Lu K. et al., 2018). Based on studies from 2006, an OH source is missing in the current chemical mechanism under low $NO_x$ conditions (Lu et al., 2012, 2013; Hofzumahaus et al., 2009). Studies performed in Wangdu (summer) and Beijing (winter) provided evidence of missing $RO_2$ sources, which could lead to a strong underestimation of the ozone production (Tan et al., 2017, 2018a). The radical observation and model comparisons highlight the uncertainty of the radical chemistry in China. Nitrous acids (HONO) were measured and constrained to the model in these studies; they are the major sources of the $OH$–$HO_2$–$RO_2$ radical system. However, the majority of HONO sources remains unclear (Su et al., 2011; Ye et al., 2016; Li X. et al., 2014). On the other hand, the large aerosol content provides a large surface for heterogeneous reactions. Radical loss on the aerosol surface could also play a role, which is not well understood due to limited information about the Chinese aerosol composition. Recently, the chlorine chemistry has gained increasing attention; it acts as a radical and particulate nitrate formation source (Tham et al., 2016; Wang et al., 2016; Wang H. et al., 2017; Wang X. et al., 2017; Wang Z. et al., 2017). However, the large variabilities in the uptake coefficient and $ClNO_2$ yield add large uncertainty to heterogeneous reactions (Xue et al., 2014; Tham et al., 2018). Although the new city clusters also suffer from air pollution, only few studies have been conducted in these regions, especially with respect to the formation of secondary pollution. Only few studies have been performed regarding the oxidation capacity in the SCB region. Chengdu was evaluated using an observational-based model; the results indicate that the radical concentration and ozone production rate are similar (Tan et al., 2018b). The volatile organic carbon (VOC) and ozone formation in Chongqing were evaluated (Su et al., 2018; Li et al., 2018b).

Radical chemistry provides insights into key processes regarding the formation of secondary pollutants. The ozone production rate can be directly determined using the oxidation rate of NO by $HO_2$ and $RO_2$. Gas phase oxidation produces semi- and/or low volatile compounds, which are important precursors for particle formation. Sulfate, nitrate, and SOA (secondary organic aerosol) are dominant contributors to

particles during heavy polluted episodes (haze events), all of which can be produced during OH-initiated oxidation processes.

In this study, we present measurements for four megacities: Beijing, Shanghai, Guangzhou, and Chongqing. The aim of this study is to illustrate atmospheric oxidation processes in urban areas. The AOC can be defined as the sum of the respective oxidation rates of trace gases (VOCs, CO, and $NO_x$) by oxidants (OH, $O_3$, and $NO_3$; Geyer et al., 2001). Given the relative importance of OH oxidation during daytime, the AOC was restricted to OH oxidation in this study. Typical photochemical polluted seasons were selected as observation periods to explore the photochemistry and secondary pollution formation. An observational-based model was used to explore the oxidation capacity from the aspect of radical chemistry. This study aims to provide insights into the formation of secondary pollution in Chinese megacities. We compared the results from different studies to illustrate common features of AOCs of megacities. Two sets of questions were addressed in this study: 1) What is the oxidation capacity in these megacities and which source(s) sustain it? In addition to foreign countries, the city centers were compared with suburban and rural locations; and 2) What is the formation rate of secondary pollutants rate, for example, ozone and nitrate, and what limits the formation of secondary pollution? The diagnosis of the AOC and formation of secondary pollution could provide fundamental information for future air pollution control in China.

## 2. Methods

### 2.1 Measurement sites

This study presents measurements performed in four Chinese megacities: Beijing, Shanghai, Guangzhou, and Chongqing. These cities are located in the highly polluted region (Fig. 1). Beijing, Shanghai, and Guangzhou are the main cities in the NCP, YRD, and PRD, which represent the most developed regions in China. Chongqing is one of the biggest cities in Southwest China (population: 30 million), representing Chinese developing city clusters.

Details about the field campaigns are summarized in Table 1. All measurement sites are located within downtown areas to represent conditions in the city center. Beijing is located on the northern edge of the NCP and is the northernmost station (39.9°N). The campaigns were mainly conducted in summer to represent the most active photochemical season. The Beijing campaign took place in mid-summer (July)

and during the strongest solar radiation input. The observation periods (August) and latitudes of Shanghai (31.1°N) and Chongqing (29.6°N) are similar, with comparable solar input levels. Given the interaction of the synoptic flow pattern, the atmospheric pollution is expected to be most serious in the PRD in autumn (Zhang et al., 2007; Li J. et al., 2014). The measurement in Guangzhou was performed in late October to represent the photochemically polluted period. However, the latitude of Guangzhou is the lowest (23.1°N), which partly compensates for the seasonal effects. The maximum of 1-h-averaged $O_3$ concentrations was higher than 100 ppbv in Shanghai, Beijing, and Guangzhou but only 79 ppbv in Chongqing, which demonstrates the separation between the two classes of cities.

## 2.2 Instrumentation

Similar instrumentation was deployed at all sites. All Thermo Fisher Scientific (shorten as Thermo) instruments were carefully maintained and calibrated during the campaigns. A brief description of the measurement techniques is presented here. Ozone was measured with the UV absorption method using the Thermo $O_3$ analyzer (Model 49i). The $NO_2$ measurement was performed using NO chemiluminescence and chemical conversion with a molybdenum convertor. This conversion method is known to show interference with $NO_z$ species, which can be converted to NO. Therefore, it should be noted that the $NO_2$ measurement presented in this study might be positively biased due to the ambient $NO_2$ concentrations. The CO content was measured via infrared absorption spectroscopy using a Thermo instrument (Model 20). The performances of different instruments are summarized in Table S1. The VOC measurements (including 55 organic species) were performed using commercial instrumentation including a gas chromatograph (GC) equipped with a mass spectrometer (MS) and flame ionization detector (FID). The air sample was drawn into two parallel channels for enrichment by cooling before analysis (Wang et al., 2014). The VOCs measurements include C2–C11 alkanes, C2–C6 alkenes, and C6–C10 aromatics (Table S2). The photolysis frequencies were measured using a spectrum radiometer. Meteorological parameters, such as the ambient temperature, pressure, and relative humidity, were simultaneously measured.

## 2.3 Model description

A box model based on the Regional Atmospheric Chemical Mechanism version 2 (Goliff et al., 2013) was used to simulate the concentrations of OH, $HO_2$, and $RO_2$ radicals and other unmeasured secondary

species. Newly proposed isoprene mechanisms were also incorporated (Peeters et al., 2014; Fuchs et al., 2013). The model was constrained to observations of photolysis frequencies, long-lived trace gases (NO, $NO_2$, $O_3$, CO, $C_2$–$C_{12}$ VOCs), and meteorological parameters. Nitrous acid (HONO) was not measured during these campaigns. Therefore, it was fixed to 2% of the observed $NO_2$ concentrations because a good correlation with a constant ratio of 0.02 was determined between HONO and $NO_2$ in different field studies (Elshorbany et al., 2012). The uncertainty of such a parameterization is discussed in Section 4.2. The measured, modeled, and parameterized data are summarized in Table S3. The uncertainty of the model calculations depends on the model constraints and reaction rate constants. Considering the uncertainties of both the measurements and kinetic rate constants, the uncertainty of the model calculations is approximately 40% (Tan et al., 2017).

## 3. Results

### 3.1 Measurement overview

The mean diurnal profiles of the measured ambient temperature; $j(O^1D)$; and CO, $O_3$ ($O_x = O_3 + NO_2$), $NO_x (= NO + NO_2)$, and AVOC (anthropogenic volatile organic compound) concentrations are shown in Fig. 2 (the time series are shown in Figs S1–S4). The ambient temperature in Beijing, Shanghai, and Chongqing is relatively similar but lower in Guangzhou because the campaign was conducted later in the year. Similarly, the photolysis frequencies are smaller in Guangzhou. However, the $j(O^1D)$ is the highest in Beijing; the values for Shanghai and Chongqing are comparable. The diurnal maximum $O_3$ concentrations are the highest in Shanghai (80 ppbv), followed by Beijing (72 ppbv), Guangzhou (65 ppbv), and Chongqing (56 ppbv). The diurnal peak of $O_3$ appears at 3–4 pm LT in Beijing, Guangzhou, and Chongqing. In Shanghai, the $O_3$ peaks occurs at 1 pm LT due to the fast $O_3$ increase in the morning. The ozone concentrations observed in Beijing, Shanghai, and Guangzhou during the measurement period exceed the Chinese National Air Quality Standard Grade II (99.3 ppbv; Table 1).

When a measurement site is close to $NO_x$ emission sources, part of the $O_3$ is converted to $NO_2$ upon reacting with freshly emitted NO. Although $O_3$ is regenerated within a few minutes to half an hour after the photolysis of $NO_2$, $O_3$ is temporarily stored in the form of $NO_2$. Therefore, $O_x$, the sum of $O_3$ and $NO_2$, is a better metric to describe ozone pollution in urban areas. The $O_x$ concentrations are shown in Fig. 2 (broken lines). The mean $O_x$ diurnal profiles in Beijing, Shanghai, and Guangzhou show

maximum values of ~90 ppbv (1-hour resolution), indicating that the ozone pollution in these cities during the measurement period is comparable. The maximum of the diurnal average in Chongqing is 66 ppbv. The ozone pollution is serious in autumn in Guangzhou due to a unique synoptic system including the surface high-pressure system, hurricane movement, and sea–land breeze (Fan et al., 2008). In Shanghai, the synoptic weather is crucial to pollution accumulation and ozone concentrations are reduced in August and September due to the cleaning effect of the summer monsoon (Dufour et al., 2010; Geng et al., 2015).

Given the relatively short periods of these campaigns, the representativeness of the measurements is a concern. We compared the observations from these intensive campaigns to routine measurements obtained at environmental monitoring stations operated by the Chinese Environmental Protection Agency (Fig. S5). The mean diurnal profiles of $O_3$ and $O_x$ obtained at all sites are comparable to the highest monthly average diurnal profiles for the same city (bias < 20%). The relatively small $O_3$ and $O_x$ concentrations observed in Chongqing compared with that of other cities (Fig. 2) are consistent with the observations made at environmental monitor stations (Fig. S5). This suggests that the ozone pollution is less severe in Chongqing compared with that in other megacities in eastern China.

The ozone precursors, $NO_x$, and AVOCs are shown in Fig. 2. All $NO_x$ concentrations show a typical diurnal profile with a minimum in the afternoon. In the morning, the peak is caused by transportation emission during the rush hour. In Shanghai, the nighttime $NO_x$ concentrations decrease after sunset but increase after midnight, which anticorrelates with $O_3$. The AVOC concentrations show diurnal profiles similar to those of $NO_x$, which suggests that both AVOCs and NOx originate from the same source, that is, traffic emission, and/or are manipulated by the same factor, for example, boundary layer development. The CO is also a precursor of ozone. The diurnal profiles of CO are almost flat due to their long lifetime compared with that of OH. A small peak appears in the morning rush hour due to poor dilution conditions and enhanced emissions from transportation.

## 3.2 OH reactivity and composition

The OH reactivity ($k_{OH}$) represents the pseudo-first-order reaction rate constant of the OH radical. It is a measure of the sum of sink terms based on the OH radical reactants $X_i$, which depends on their ambient concentration $[X_i]$ and rate coefficient with the OH radical. Mathematically, $k_{OH}$ equals the inverse of the

ambient OH radical lifetime. The use of the OH reactivity is of importance to understand the OH consumption potential.

The mean diurnal profiles of the OH reactivity calculated using a box model are presented in Fig. 3 including the contribution from CO, $NO_x$, VOCs, and model-generated secondary species. In general, the OH reactivity is the lowest in the afternoon and the highest in the morning rush hour due to the change in the dilution conditions throughout the day.

In this study, the OH reactivity modeled for Guangzhou is the highest (20–30 $s^{-1}$) among all cities (Fig. 3), indicating the strong influence of anthropogenic emissions in Guangzhou. The OH reactivities of Beijing and Chongqing are comparable, ranging from 15 to 25 $s^{-1}$. The OH reactivity of Shanghai is the lowest (<15 $s^{-1}$) due to the small contribution from CO and $NO_x$.

The OH reactivity measurements performed in previous field campaigns in NCP and PRD showed large seasonal and spatial variations. In Beijing, $k_{OH}$ measurements were conducted at urban and suburban sites during summer (Lu et al., 2013; Williams et al., 2016; Yang et al., 2017). The OH reactivity is on average in the range of 10 to 30 $s^{-1}$ and shows a large daily variation due to meteorology changes. In the PRD, the OH reactivity was measured at a rural site (Backgarden) in summer and at a suburban site (Heshan) in autumn, both of which are close to the megacity Guangzhou (<100 km). In Backgarden, the average OH reactivity was in the range of 20 to 50 $s^{-1}$, with a large contribution from isoprene (Lou et al., 2010). In Heshan, downwind of Guangzhou, the OH reactivity was dominated by anthropogenic species. However, the average $k_{OH}$ of the two sites was comparable (Tan et al., 2018c). In Chengdu, another big city in southwestern China, the modeled OH reactivity was in the range of 15 to 30 $s^{-1}$ at three urban sites; the large OH reactivity was contributed by alkenes derived from the petrochemical industry (Tan et al., 2018b).

The contributions of different OH reactivities are shown in Fig. 4. The contribution of the OH reactivity was relatively similar in all cities, which is a typical fingerprint of anthropogenic emission. The $NO_x$ is the most important OH reactant, contributing 28%–35% to the total reactivity. In total, the inorganic species (CO and $NO_x$) contribute more than 50% of the reactivity. The measured AVOCs contribute between 14% and 26% to the total OH reactivity. The AVOC contribution and absolute reactivity (2.3 $s^{-1}$) in Beijing are the lowest. The contribution and absolute reactivity of AVOCs (6 $s^{-1}$ compared to 2–3 $s^{-1}$ in other cities) in Guangzhou are the highest. However, the isoprene reactivity is relatively small in Guangzhou (0.4 $s^{-1}$) due to the observation season. In comparison, the isoprene reactivity reaches up to

1 s$^{-1}$ in Beijing and Chongqing but is negligible in Shanghai. Small isoprene concentrations were also reported in another study (Geng et al., 2011).

**3.3 Modeled OH–HO$_2$–RO$_2$ radical concentrations**

The OH, HO$_2$, and RO$_2$ concentrations were derived from box model calculations (Figs S1–S4). The mean diurnal profiles are shown in Fig. 5. The OH concentrations modeled for all campaigns show distinct diurnal variations. The peak of the OH mean diurnal profile is the highest in Beijing and Shanghai (7 × 10$^6$ cm$^{-3}$). The peak is 4 × 10$^6$ cm$^{-3}$ in Chongqing and further decreases to 2 × 10$^6$ cm$^{-3}$ in Guangzhou. The relative change of the OH maximum is consistent with the observed photolysis frequencies (Fig. 2). In fact, the modeled OH concentrations show a good correlation with the observed j(O$^1$D) (R = 0.83, 0.93, 0.87, and 0.87 for Beijing, Shanghai, Guangzhou, and Chongqing, respectively) because the OH chemistry strongly relies on solar radiation input. The OH maximum ranking is consistent with that of j(O$^1$D). The j(O$^1$D)–OH correlation slope is the largest in Shanghai (3.0 × 10$^{11}$ cm$^{-3}$s$^{-1}$ compared with 2.0 × 10$^{11}$ cm$^{-3}$s$^{-1}$ in the other cities) because the OH reactivity is lower and the OH lifetime becomes longer. A good correlation between OH and j(O$^1$D) with a correlation coefficient >0.8 was observed in other field campaigns conducted in China (Tan et al., 2017; Rohrer et al., 2014; Lu et al., 2012, 2013). However, the correlation slopes were ~(4.5 ± 0.5) × 10$^{11}$ cm$^{-3}$s$^{-1}$, that is, on average two times larger than the results in this study. The different slopes could be due to the different chemical regimes at each location. A larger factor was determined at suburban and rural sites, where the air masses were more oxidized. The slope is comparable to the results obtained in urban environments, where the slope was in the range of 2 to 4 × 10$^{11}$ cm$^{-3}$s$^{-1}$ (Holland et al., 2003; Michoud et al., 2012; Griffith et al., 2016; Whalley et al., 2018).

The peroxy radical concentrations determined in different campaigns vary depending on the chemical conditions. All cases show a general feature, that is, the peroxy radical concentrations are suppressed in the morning. They gradually increase later and reach a peak at approximately 2 pm LT (two hours after the maximum solar radiation) because of the suppression by high NO. The peroxy radical concentrations are highest in Chongqing, with maximum mean diurnal profiles of 5 × 10$^8$ cm$^{-3}$ for HO$_2$ and 7 × 10$^8$ cm$^{-3}$ for RO$_2$. The HO$_2$ and RO$_2$ concentrations in the other cities are comparable despite the differences in the solar radiation and chemical conditions. In Chongqing, the relatively large

VOC/NO$_x$ ratio leads to the highest peroxy radical concentration in the model, which reflects the efficient radical recycling and ozone production.

Previous field campaigns in China showed that the OH concentrations might be underestimated under low NO$_x$ conditions (Tan et al., 2017, 2018a; Fuchs et al., 2017; Rohrer et al., 2014; Lu et al., 2012, 2013; Hofzumahaus et al., 2009). In this study, the NO$_x$ concentrations are moderate or high, where the model is capable of reproducing the OH concentrations relatively well (Rohrer et al., 2014). The prominent feature in the high NO$_x$ regime is the underestimation of the HO$_2$ and RO$_2$ concentrations (Tan et al., 2017, 2018a), which was also observed at other urban sites outside of China (Griffith et al., 2016; Whalley et al., 2018; Kanaya et al., 2007; Dusanter et al., 2009; Shirley et al., 2006; Brune et al., 2016; Ren et al., 2013). This indicates a common flaw in current chemical mechanisms. Such a model inadequacy can lead to the underestimation of the local ozone production. The explanation of the model underestimation is out of the scope of this study, but its possible impact is discussed in Section 4.3.

## 4. Discussion

### 4.1 VOC compositions and ozone production efficiency

The relative contributions of grouped VOCs are shown in Fig. 6. Anthropogenic VOCs are usually more important than the biogenic ones. Isoprene accounts for 15% of the measured VOC reactivity in Beijing and Chongqing. The shares of different VOCs groups are comparable, with a slightly different portion of alkanes and alkenes. Aromatics are the most important VOC group in Guangzhou, accounting for ~50% of the measured VOC reactivity, which is related to vehicle emissions and industries that produce VOC-related products (Zheng et al., 2009).

The OH reactivity concept is useful to estimate ozone production for VOCs because it describes the VOC degradation rate initiated by OH oxidation, which leads to net ozone production in the presence of NO and sunlight (see Section 4.1). On the other hand, the ozone formation potentials (OFPs) are used to describe the theoretical ozone production maximum. This metric indicates the temporal Lagrangian evolution of the O$_3$ production potential within a mixture of air that undergoes full oxidation. The OFP of individual VOCs is calculated by the product of the measured VOC concentrations and its MIR (maximum incremental reactivity) value (Carter, 2009), which are summed up later according to the

VOC classification. Aromatics become the dominant species in the OFPs due to their large carbon numbers and high MIR values. In Guangzhou, aromatics contribute up to 70% of the share of OFPs, followed by Shanghai (55%), Beijing (43%), and Chongqing (42%).

The mixing ratios of different VOCs are shown for comparison (Fig. 6). Note that alkanes (including ethyne) account for a large fraction of the mixing ratio; their contribution to the OFPs and OH reactivity is small. Therefore, they are not important for the ozone formation and radical chemistry. This highlights the importance of the concept of OH reactivity and OFPs, which can be used to accurately describe photochemical processes.

The top ten VOCs contributing to the OH reactivity are summarized in Table 2 in order of importance based on the average OH reactivity in four cities. Among all, propene is the most important OH reactivity contributor ($\sim$0.4–0.6 s$^{-1}$; Table 2). Small VOCs (propene, ethane, and ethene) are relatively important with respect to the OH reactivity. Nine of the top ten VOCs are alkenes and aromatics (except for ethane). In Guangzhou, xylene (m,p-, and o-) and toluene are also important OH reactants, which is consistent with an earlier inventory study (Zheng et al., 2009). The diurnal profiles are shown in Fig. S6. The observed anthropogenic VOC concentrations show a typical diurnal profile that increases during the night and decreases during the afternoon. One exception is the Shanghai site where the mean diurnal profiles of propene and 1,2,4-trimethybenzene are flat, while the styrene profile shows a peak around noontime, indicating a unique VOC emission feature at that site.

The OVOC (oxygenated volatile organic compound) concentrations were simulated using the box model. The modeled HCHO concentrations range from 3 to 8 ppbv (Fig. S7) and are consistent with previous studies in these regions (Zhang et al., 2012; Song et al., 2018; Chen et al., 2016; Tang et al., 2009). The modeled acetaldehyde concentrations range from 2 to 3 ppbv in Beijing, Shanghai, and Chongqing but are 1 ppbv larger in Guangzhou because of the larger contribution of aromatic VOCs, which produce acetaldehyde during OH degradation.

## 4.2 OH–HO$_2$–RO$_2$ radical budget analysis

All radical reactions are classified into four groups (initiation, termination, propagation, and thermal equilibrium with reservoir species). The reaction turnover rate illustrates important processes in the RO$_x$ radical reaction framework. The initiation and termination rates are shown in Fig. 7. The

following radical budget analysis will focus on the daytime conditions (6 am to 6 pm LT) unless otherwise noted.

The dominant radical sources are photolysis reactions involving HONO, $O_3$, HCHO, and other carbonyl compounds. The photolysis of HONO and $O_3$ (producing $O^1D$ and followed by a $H_2O$ reaction) produces OH radicals, which contribute 33%–45% to the total primary sources, $P(RO_x)$, among all campaigns. The HCHO photolysis produces $HO_2$ [14%–33% of $P(RO_x)$], while the other carbonyl compounds generate $RO_2$ radicals [3%–6% of $P(RO_x)$]. Therefore, photolysis reactions dominate the primary radical sources during daytime (58%–86%). In contrast, alkene ozonolysis is the dominant radical source during nighttime; the yields of OH, $HO_2$, and $RO_2$ radicals depend on individual alkenes. The maximum $P(RO_x)$ mean diurnal profile is largest in Beijing (5 ppbv/h), followed by Shanghai (4.6 ppbv/h) and Chongqing (4.3 ppbv/h). The average daytime $P(RO_x)$ is smaller in Shanghai due to the narrower photolysis frequencies (Fig. 2) and shorter photolysis reaction time. The primary radical source is the smallest in Guangzhou (3.2 ppbv/h) because of the later observation period in the year. However, alkene ozonolysis significantly contributes to the radical sources in Guangzhou (43% of the total primary sources under daytime conditions), which could attribute to the higher abundance of alkenes due to the special emission inventory (see Section 3.2).

The ozonolysis reactions mainly involve trans-2-butene in Beijing (55%), Guangzhou (42%), and Chongqing (39%), with concentrations ranging from 0.1 to 0.3 ppbv (Fig. S6). Although trans-2-butene is only the eighth important VOC with respect to the OH reaction (Table 2), it becomes the most important $O_3$ reactant producing $RO_x$ radicals due to its fast reaction rate with $O_3$ [$1.9 \times 10^{-16}$ $cm^{-3}s^{-1}$ compared with $1.0 \times 10^{-17}$ $cm^{-3}s^{-1}$ for propene; rate constant derived from MCM3.3.1 (http://mcm.leeds.ac.uk/MCMv3.3.1/home.htt)]. In Shanghai, propene is the most important alkene with respect to the $O_3$ reaction; it accounts for ~42% of the total alkene ozonolysis reactions. In fact, the relatively high contribution of alkene ozonolysis to primary $RO_x$ radical sources could be one of the important characteristics of primary $RO_x$ radical sources in Chinese megacities. The importance of alkene ozonolysis was also observed in Santiago, Chile (Elshorbany et al., 2009), and Essex, UK (Emmerson et al., 2007), where alkene ozonolysis contributed ~20% to the total primary radical production.

Radical termination can be divided into two groups, that is, nitrogen-containing compounds, including HONO, $HNO_3$, $RONO_2$, and PAN-type (peroxy acyl nitrate) species ($L_N$). The other pathway leads to

peroxide formation due to the combination of two peroxy radicals ($L_H$). The ratio between $L_N$ and $L_H$ depends on the $NO_x$ concentrations. In urban environments, the factor limiting radical propagation is the abundance of VOCs. In our case, the radical termination is dominated by $L_N$ (>70%). Nitric acid formation is the major contributor to radical termination in all cities (>50%). In Chongqing, the peroxide formation path contributes 26% to radical termination. The ratio increases to 32% during the afternoon due to the higher VOC/$NO_x$ ratio. The formation of net PAN-type species based on radical loss becomes relatively important in Guangzhou (~20%) due to the lower temperature (Fig. 2). It has been reported that on average 25% of the radical is lost via the formation of PAN-type species in Beijing during winter (Tan et al., 2018a). In addition, the formation of PAN-type species becomes important in urban areas. For example, it contributes 30% to the total radical loss in downtown London (Whalley et al., 2018).

A comparison of the four cities is clearly shown in Fig. 8. The HONO photolysis is the dominant OH source in all cities, except for Shanghai. The $O_3$ photolysis is more important than HONO photolysis in Shanghai, contributing 55% to the total primary OH sources and 23% to the total radical sources. In all cities, the primary production of $HO_2$ is comparable to that of OH, which is mainly contributed by HCHO photolysis and alkene ozonolysis. These results are consistent with the model calculation performed for Beijing (Yang et al., 2018) and Hong Kong (Xue et al., 2016). The importance of HONO and HCHO photolysis for primary radical production was also reported for suburban and rural environments (Lu et al., 2012, 2013; Tan et al., 2017). In the base model scenario, HONO was scaled to $NO_2$ measurements; the uncertainty of this assumption is further discussed below. In the base model, the HONO concentrations are scaled to the observed $NO_x$ concentration using a scaling factor of 0.02 (Elshorbany et al., 2012). In this study, we used this scaling factor between HONO and $NO_x$ to simplify the discussion of unknown HONO sources. In the original RACM2 model, only homogenous sources are included, that is, OH + NO $\rightarrow$ HONO, which does not support the high daytime HONO concentrations and therefore leads to a strong underestimation of the OH concentrations (Su et al., 2011; Yang et al., 2014; Tong et al., 2016; Ye et al., 2016; Li X. et al., 2012, 2014). Although the HONO to $NO_2$ ratio is relative robust and constant, as reported in other field campaigns (Elshorbany et al., 2012), such a simple parameterization could increase the uncertainty of our model calculation. To further investigate the uncertainty based on this simple parameterization, the scaling factor was varied from 0.015 to 0.03. The modeled OH concentrations change by less than 10% if the HONO scaling factors

change by 50% (Fig. S8). In addition, the modeled $HO_2$ and $RO_2$ concentrations are relatively stable at different HONO scaling factors. The different scaling factors also affect model-generated species such as HCHO (Table S4). In fact, higher HONO concentrations lead to more active photochemical reactions and a greater HCHO production. In return, higher HCHO concentrations could further enhance the photochemistry due to more radical photolytic sources. Therefore, the higher (lower) modeled radical concentrations based on the increase (reduction) of HONO scaling factors are also affected by the corresponding change of the modeled HCHO concentrations. This demonstrates the nonlinearity of the photochemical system. Nevertheless, the parameterized HONO concentrations range from 0.3 to 0.6 ppbv during the daytime (Table S4), which is consistent with previous *in situ* measurements in urban areas (Lu et al., 2013; Li et al., 2010; Ren et al., 2003; Kanaya et al., 2007). To evaluate the impact of a missing HONO source on the radical chemistry, we switched off the scaling between HONO and $NO_x$ in the sensitivity test (Fig. S8). The results show that the OH concentrations decrease by ~20% if only a homogenous source is considered. The modeled $HO_2$ and $RO_2$ concentrations also decrease by 15%–20% (Table S4).

The primary $RO_2$ source strength is in the range of 0.2 to 0.3 ppbv/h, which is mainly contributed by alkene ozonolysis and OVOC photolysis (Fig. 8). In this study, OVOC photolysis mainly involves carbonyl-containing compounds (e.g., acetaldehyde, aldehydes with carbon numbers larger than three), which are generated by the box model. The modeled acetaldehyde concentrations range from 2 to 4 ppbv (Fig. S7), consistent with observations in Beijing (Chen et al., 2016) and Hong Kong (Lyu et al., 2016). The photolysis rate of carbonyl-containing compounds (except HCHO) is approximately one-third to one-quarter of the HCHO photolysis rate. For comparison, this ratio can be close to or even higher than one based on other urban studies (Ren et al., 2013; Volkamer et al., 2010; Emmerson et al., 2007; Michoud et al., 2012; Whalley et al., 2018; Xue et al., 2016). In contrast, a relatively small contribution from the photolysis of carbonyl-containing compounds other than HCHO was reported for urban and suburban sites in Hong Kong (Lyu et al., 2016), where the acetaldehyde concentrations ranged from ~1 to 2 ppbv, comparable to our model simulation. Such a large discrepancy in the contribution of photolysis of other OVOCs to the radical production highlights the importance in measuring these radical precursors in future studies.

The OH reacts with VOCs or CO and produces peroxy radicals. The peroxy radicals react with NO to produce $NO_2$, leading the net production of $O_3$ in the presence of sunlight (see Section 4.3). Nitric acid

can be formed from the reaction between $NO_2$ and OH, which is an important precursor of fine particles (see Section 4.4). Therefore, efficient radical propagation facilitates the formation of secondary pollution.

In addition, the radical propagation between OH, $HO_2$, $RO_2$, and $RCO_3$ is also shown in Fig. 8. In RACM2, $RCO_3$ is mainly produced by the reaction between OH radicals and aldehydes. The $RCO_3$ is separated from the sum of the $RO_2$ family and explicitly shown in the radical propagation because $RCO_3$ reacts with NO and is converted to $RO_2$. In the sum of $RO_2$ concentrations, $RCO_3$ is considered as a subgroup of $RO_2$ radicals (e.g., Fig. 5).

The conversion from OH to $RO_2$ is slightly faster than that from OH to $HO_2$. In addition, the rate of the conversion from OH to $RCO_3$ is approximately one-third of that from OH to $RO_2$. In the presence of NO, OH is regenerated from peroxy radicals and ozone is produced during the same process. The flow rate from $RO_2$ to $HO_2$ is less than half of the flow rate from $HO_2$ to OH. When considering the contribution from $RCO_3 + NO$, $RO_2$ (+$RCO_3$) contributes less than half of the $P(O_3)$. Surprisingly, the $HO_2$ contribution to the total ozone production is constant (63%) in all cases. In this study, the ratio between $P(O_3)$ and $k_{voc} \times [OH]$ is rather constant, ranging from 1.5 to 1.6. Therefore, the robust relation between OH oxidation and ozone production indicates that future estimations of the ozone production rate using the reaction rate of OH + VOC (CO) are justified. However, the nature of such a robust relation needs to be investigated before real application.

## 4.3 Ozone production and sensitivity

### 4.3.1 Local ozone production

Ozone is generated during $NO_2$ photolysis, which simultaneously produces NO. The removal of NO without consuming $O_3$ leads to net ozone production. In the photochemical system, peroxy radicals ($HO_2$ and $RO_2$) are the major NO consumers in all photochemical reactions.

In this study, the ozone formation rate $F(O_3)$ was calculated from the rate of NO oxidation by $HO_2$ and $RO_2$ radicals, as denoted in Eq. (1).

$$F(O_3) = k_{HO_2+NO} \times HO_2 \times NO + k_{(RO_2+NO)eff} \times RO_2 \times NO \qquad (1)$$

The chemical loss of $O_3$ is due to $O_3$ photolysis (production of $O^1D$, followed by $H_2O$ reaction) and its reactions with alkenes, OH, and $HO_2$ [Eq. (3)]. Because $NO_2$ can be regarded as a reservoir species of

$O_3$, the reaction between $NO_2$ and OH is also considered to reflect chemical ozone loss. In fact, the use of $O_x$ is helpful to avoid the interruption of NO titration due to fresh emission, which is conservative to describe the ozone concentration change due to photochemical reactions (Liu, 1977).

$$D(O_3) =$$

$$[O^1D][H_2O] + (k_{O_3+OH}[OH] + k_{O_3+HO_2}[HO_2] + k_{O_3+alkenes}[alkenes])[O_3] + k_{OH+NO_2}[OH][NO_2]$$

$$(2)$$

Therefore, the net ozone production rate $P(O_3)$ is determined by the difference between Eq. (1) and (2). The mean diurnal profiles of $P(O_3)$ are shown in Fig. 9. The ozone production rate is the highest in Beijing and Shanghai, with an average diurnal maximum reaching 19 ppbv/h. Although the peroxy radical concentrations are the highest in Chongqing, the ozone production rate only shows a broad peak at 13 ppbv/h. The duration of the ozone production differs; the longest duration is observed in Beijing (13 hrs). Therefore, the daily integrated ozone production rate is the largest in Beijing (136 ppbv). For comparison, the integrated ozone production rate is 92, 40, and 105 ppbv in Shanghai, Guangzhou, and Chongqing, respectively.

The $O_x$ concentration change depends on both the local production and physical processes [Eq. (3)]. The $R(O_3)$ represents the combined effect of all physical processes including horizontal transportation, vertical mixing, and deposition.

$$\frac{d(O_x)}{d(t)} = P(O_3) + R(O_3) \qquad (3)$$

As shown in Fig. 9, the $O_x$ concentration changes are derived from the derivative of the observed $O_x$ concentrations. In Beijing, Guangzhou, and Chongqing, the $d(O_x)/dt$ shows a similar increase in the morning starting at 6 am LT (Fig. 9). In Shanghai, a positive derivative is observed at 5 am LT, one hour earlier than in the other three cities. Also, the increase rate is the fastest in Shanghai during the morning hours. However, the $d(O_x)/dt$ sharply changes from positive to negative at noon in Shanghai, while it remains positive until 4 pm LT in the other three cities. The difference in the $d(O_x)/dt$ results in an early $O_x$ peak (around noon time) in Shanghai.

The difference between $d(O_x)/dt$ and $P(O_3)$ is shown in the bottom panel of Fig. 9, which denotes the local transportation of $O_x$ (positive: inflow; negative: outflow). The local production rate is larger than the $O_x$ concentration increase in Beijing and Chongqing. This suggests that the photochemically produced $O_x$ at the measurement site is transported to downwind regions. In Shanghai and Guangzhou,

both positive and negative values are observed during daytime, indicating that the local ozone budget changes from import to export. In the morning, the increase in the $O_x$ concentration is larger than the local ozone production rate, which is supported by additional $O_x$ import from the volume outside. The $O_x$ is most likely entrained from the air because the rising boundary layer mixes in the air mass from the residual layer, which maintains the high $O_x$ load that was produced on the previous day and then isolated from the surface layer. The fast $O_x$ concentration increase before 8 am LT is mainly caused by transportation, especially in Shanghai, given the relatively small local production rate. The $O_x$ import stops at approximately 10 am LT and the surface layer becomes the net $O_x$ source region. The $O_x$ import stops later (at approximately 2 pm LT) in Guangzhou. The $O_x$ transport to downwind areas [negative $R(O_3)$] occurs during the afternoon in Beijing, Shanghai, and Chongqing and after sunset in Guangzhou, which suggests that the city centers are important for the $O_x$ formation on a regional scale.

As mentioned in Section 3.3, the current model could have flaws under high-$NO_x$ conditions, which might lead to the underestimation of the peroxy radical concentrations and thus the local ozone production in China (Tan et al., 2017, 2018d; Griffith et al., 2016; Whalley et al., 2018; Kanaya et al., 2007; Dusanter et al., 2009; Shirley et al., 2006; Brune et al., 2016; Ren et al., 2013). However, quantitative estimation is not possible due to the absence of *in situ* radical measurements. To our knowledge, no field campaigns have been conducted to perform *in situ* radical measurements in city center areas in China. However, based on a field campaign in the downwind area of Beijing (Yufa), the local ozone production rate was underestimated due to the underestimation of the $HO_2$ concentrations (Lu et al., 2010). The results of another field campaign at a rural site in the NCP also showed model underestimation of $P(O_3)$ by 20 ppbv per day compared with a daily integrated ozone production of 110 ppbv derived from the measured $HO_2$ and $RO_2$ (Tan et al., 2017). Therefore, the ozone production rate derived from model calculations in this study should be considered to be a lower limit. Nevertheless, the underestimation of the peroxy radical concentration will not affect the $O_3$–$NO_x$–VOC sensitivity diagnosis (Tan et al., 2018d).

**4.3.2 $O_3$–$NO_x$–VOC sensitivity**

The OH–$HO_2$–$RO_2$ radical budget is useful for the diagnosis of the $O_3$–$NO_x$–VOC sensitivity, as discussed in Section 3.3.2. In this section, we use the ratio of the nitrate formation rate ($L_N$) to the total radical production/termination rate (Q), known as $L_N$/Q ratio, to evaluate the ozone production

sensitivity, as suggested by Kleinman et al. (1997). The threshold of the $L_N/Q$ ratio is 0.5. When $L_N/Q$ is greater than 0.5, the radical termination is outweighed by the nitrate formation, which indicates that the ozone production is limited by the VOC abundance. On the other hand, when $L_N/Q$ is less than 0.5, peroxy radical self-combination dominates the radical termination, indicating that the ozone production is controlled by $NO_x$. The radical budget analysis shows that $L_N$ contributes more than 70% to the radical termination in all cities. At a $L_N/Q$ ratio above 0.5, the ozone production is in the VOC-limited regime, which persists in all cities in this study.

The Relative Incremental Reactivity (RIR) method can also be used to evaluate the $O_3$–$NO_x$–VOC sensitivity. The RIR is a useful metric for the ozone sensitivity to individual precursors. The model input parameters are altered to a certain extent and the corresponding change in ozone concentration is compared and summarized to reveal the $O_3$–$NO_x$–VOC sensitivity. The RIR can be calculated using Eq. (5).

$$RIR(X) = \frac{\Delta O_3(X)/O_3}{\Delta C(X)/C(X)} \qquad (4)$$

In Eq. (4), X represents a set of primary pollutants and $O_3$ represents the $O_3$ concentrations modeled for the base case. The term $\Delta C(X)/C(X)$ represents the relative change in primary pollutants in one of the sensitivity tests. As a result, the relative change in the modeled ozone concentrations is given by $\Delta O_3(X)/O_3$.

The RIR values were calculated for $NO_x$, AVOC, NVOC, and CO, respectively (Fig. 10). The AVOC has the highest RIR (>1 %/%) in all cities because the ozone production is limited to the VOC abundance in urban areas. In comparison, the RIRs of NVOC and CO are small (<0.2%/%), which demonstrates that isoprene and CO are unimportant ozone precursors in all cities. The only component of NVOC is isoprene, which is on average 0.3 ppbv in Beijing, 0.1 ppbv in Guangzhou, and 0.4 ppbv in Chongqing but negligible in Shanghai (below the detection limit, Table S3). The RIR values for $NO_x$ are negative, indicating that the ozone production is in the $NO_x$-titration regime. A slight reduction of $NO_x$ could lead to an increase in the $O_x$ concentration within the $NO_x$-titration regime. This can be explained with the radical budget based on which the $OH + NO_2$ reaction rate is a dominant part of the radical termination in all cities (Fig. 7). If the radical termination is reduced, the $OH$–$HO_2$–$RO_2$ radical propagation becomes more efficient and thus the modeled radical concentrations increase. One should keep in mind that HONO is scaled to $NO_2$ in our base model. The reduction in $NO_2$ also leads to a reduction in HONO,

which translates into less primary radical sources. Therefore, less ozone is produced from radical recycling, which partly compensates for the titration effect. We performed another sensitivity study with free-running HONO. The results show that the negative effect becomes more significant compared with the base model scenario (Fig. S9). The reduction of radical termination by the OH + NO$_2$ reaction compensates the reduction of primary radical sources due to HONO photolysis. Therefore, a larger negative effect is observed in the RIR analysis during the sensitivity test (Fig. S9).

As discussed above, VOC emission control is critical for ozone pollution reduction. To perform accurate VOC mitigation for O$_3$ pollution control, the AVOC is further split into alkanes, alkenes, and aromatics. As shown in Section 3.2, alkenes and aromatics are the dominant VOC groups with respect to the OH reactivity. The RIR analysis also shows that reduction in alkenes and aromatics is important for ozone pollution control in these megacities (Fig. 10). Xue et al. (2014) compared observations made in downwind suburban areas of the four cities (Beijing, Shanghai, Guangzhou, and Lanzhou). The results showed that the ozone increase could be attributed to the local ozone production in Shanghai, Guangzhou, and Lanzhou (Xue et al., 2014). In addition, the ozone production was in the VOC-limited regime in Shanghai and Guangzhou and aromatics were the most important contributors. A comparison of two megacities in China (Shanghai and Tianjin) showed that the ozone production is highly variable depending on the VOC speciation at certain NO$_x$ concentrations (Ran et al., 2012). Alkenes are important ozone precursors in Tianjin, while aromatics dominate the ozone production in Shanghai. Based on a one-year measurement in Nanjing (YRD), the ozone production is in the VOC-limited regime and the Nanjing–Shanghai axis with its city clusters in between is subjected to regional photochemical pollution (Ding et al., 2013). The regional model (WRF-Chem) also showed that the ozone production is strongly VOC-limited, not only in urban but also in larger regional areas (Tie et al., 2013). The PRD region has been well studied by field campaigns, emission inventories, and regional modeling (Zhang et al., 2008a; Zheng et al., 2009; Ding et al., 2004). The results showed that both urban (Guangzhou) and downwind rural sites (Xinken) are in the VOC-limited regime; negative RIR values for NO$_x$ were observed (Zhang et al., 2008b). The ozone–NO$_x$–VOC sensitivity was reported to be VOC-limited in Chongqing and traffic emissions contribute 44% to the VOCs in an urban area (Su et al., 2018; Li J. et al., 2018).

## 4.4 Nitrate production potential

Nitric acid is one of the major products generated by the radical system under high-$NO_x$ conditions and is an important precursor of particulate nitrate ($NO_3^-$). Recently, nitrate has become a significant portion of particles in Beijing, Shanghai, and Nanjing (YRD) during summertime (Li H. et al., 2018). The gas phase nitric acid $HNO_3$ and ammonium $NH_3$ form a gas–particle partitioning equilibrium with $NH_4NO_3$ (R3), which depends on the relative humidity, temperature, and aerosol content (Seinfeld and Pandis, 2016).

$$NH_3 + HNO_3 \leftrightarrow NH_4NO_3 \qquad\qquad \text{(R3)}$$

Nitric acid is mainly produced from the reaction between NO and OH, which can be derived from the box model. The fate of nitric acid depends on the gas–particle partition, deposition, and chemical reactions. In general, the partitioning timescale is 1–2 orders smaller than that of deposition and chemical production (Morino et al., 2006; Neuman et al., 2003). Therefore, the photochemically produced nitric acid is deposited on to the aerosol if the ambient $NH_3$ is sufficient. The deposition rate is ~7 cm s$^{-1}$ (Seinfeld and Pandis, 2016), which results in a deposition timescale of 8 hrs if the boundary layer height is 2 km (typical summer value). The ammonia concentrations are usually above 10 μg/m$^3$ in urban areas in China during summer (Pan et al., 2018), which indicates ammonia-rich conditions sufficient to neutralize nitric acid.

In this study, we used the aerosol thermodynamic model (ISORROPIA) to simulate the nitrate production. The model design is explained in the Supplementary Material. Note that such a model simulation cannot be quantitative because several key parameters, such as the ammonia and nitric acid concentrations, were not measured during these campaigns. The discussion below should be considered as a qualitative estimation, indicating features that are important for the determination of the particle nitrate production. The modeled nitrate concentration and partitioning in Beijing are shown to illustrate the typical pattern of particulate nitrate formation (Fig. 11a). The maximum total nitrate concentrations are observed in the late afternoon, while the particulate nitrate shows a broad peak at night, which is mainly driven by the stronger gas-to-particle partitioning due to a higher RH. Because the deliquesce relative humidity (DRH) of $NH_4NO_3$ is ~60% in all cases, the partitioning dramatically changes when the relative humidity is above DRH (nighttime) and below DRH (daytime). Note that the nitrate formation

from $N_2O_5$ hydrolysis was not considered in this study, which could lead to a negative bias in the nitrate production calculation.

To investigate the dependence of the nitrate concentration on the nitrate production rate and ambient ammonia concentrations, the average nitrate concentrations are plotted as a function of the daily integrated nitric acid production rate and total ammonium ($NH_4^+{}_{(a)} + NH_{3(g)}$) concentration. As shown in Fig. 11b, the isopleth diagrams are split into two parts by the dashed line to represent nitrate- (upper left) and ammonium-sensitive (lower right) regimes. However, the threshold for the nitrate- and ammonium-sensitive regimes is not distinct in the small chemical range. In fact, the nitrate concentrations are sensitive to both precursors. The daily integrated nitrate production rate and average total ammonium concentrations for each city are denoted by circles (Fig. 11b). The circles are located above the ridgeline, which means that the nitrate concentrations are more sensitive to the rate of change of nitric acid production. This scenario study highlights that future mitigation of summertime particulate nitrate pollution should aim at the reduction of the photochemical production of nitric acid. For example, the reduction of $NO_x$ emissions might help to reduce the particulate nitrate pollution but may lead to enhanced ozone pollution (see Section 4.3).

**4.5 Atmospheric oxidation capacity and formation of secondary pollution**

The AOC is mainly contributed by OH radicals, which dominate the chemical removal of trace gases (e.g., CO, $NO_2$, and VOCs). The OH reactions convert primary pollutants to oxidized products (e.g., $CO_2$, $HNO_3$, and OVOCs). As shown in Fig. 12, the average daytime OH oxidation rate reaches up to 10 ppbv/h in Beijing, indicating a strong oxidation capacity (daily integrated oxidation rate > 100 ppbv). The OH oxidation rates in Shanghai and Chongqing are comparable (~5 ppbv/h); the OH oxidation rate in Guangzhou is 4 ppbv/h. In this study, the OH oxidation rate correlates with the strength of primary radical sources $P(RO_x)$. The ratio between the radical recycling rate and primary production rate indicates the efficiency of radical propagation [Eq. (6)], also known as radical chain length:

$$ChL = (k_{VOC} + k_{CO}) \times [OH]/P(RO_x), \qquad (6)$$

where $k_{VOC}$ and $k_{CO}$ represent the reactivity of VOC and CO versus the OH radical, respectively.

The radical chain length is on average 2.9 $\pm$ 0.3 in all cases, which is consistent with the results (3–5) derived from radical observations in urban areas (Kanaya et al., 2008; Ren et al., 2006; Emmerson et al., 2005).

As shown in Section 3.3.2, $P(RO_x)$ highly correlates with $j(O^1D)$ because photolysis reactions dominate the $RO_x$ primary sources. An exception was observed for Guangzhou, where the $P(RO_x)$ is by a factor of 2 smaller than that of Beijing, although $j(O^1D)$ is reduced by a factor of 3. The $P(RO_x)$ is smaller in Guangzhou because the alkene ozonolysis reactions increase to 0.7 ppbv/h (0.2–0.3 ppbv/h in other cities) and contribute nearly 50% of the primary source. The reduction of $P(RO_x)$ in Guangzhou is partly compensated.

More than 50% of the OH oxidation rate is contributed by the reaction with VOC, which produces less volatile species (OVOCs; Fig. 12). These oxidized compounds have the potential to contribute to particle formation because of their low volatility (Odum et al., 1997). In addition, the oxidation of $NO_2$ produces $HNO_3$, which contributes to the particle formation because of relatively high ambient $NH_3$ concentrations (see Section 4.2).

The $O_3$ is another important secondary pollutant generated in the OH–$HO_2$–$RO_2$ radical system. The average $P(O_3)$ is consistent with the OH oxidation rate in four cities (Fig. 12). The ratio between the ozone production and primary radical production rate is used to evaluate the ozone production efficiency (OPE). In this study, the OPE is the highest in Chongqing (3.6 on average) due to the relatively high VOC/$NO_x$ ratio. In contrast, the OPE is only 2.2 in Guangzhou and increases to 3.4 and 3.1 in Beijing and Shanghai, respectively. In comparison, the OPE ranges from 3 to 7 in other cities (Kleinman et al., 2002; Lei et al., 2008). The OPE determined in this study is low compared with that of cities in the US, which is due to the suppression of high-$NO_x$ conditions. Because the OPE generally increases with time when a plume is transported and diluted (Kleinman et al., 2002), the ozone production is more efficient in suburban areas of the megacities.

Finally, the radical inner recycling rate is on average 4–5 times larger than the radical initiation (termination), which demonstrates that the formation rate of secondary pollutants (e.g., ozone and nitric acid) is enhanced by efficient radical recycling.

**5. Summary and conclusion**

Secondary pollution has increased over the past decade in Chinese cities despite the reduction of primary pollution. Ozone and fine particle precursors (e.g., $H_2SO_4$, $HNO_3$, and ELVOCs) are generated as a result of radical reactions. The AOC is the basis for the formation of secondary pollutants. In this study, we present observations of radical precursors in four Chinese megacities during photochemically polluted seasons: Beijing (July), Shanghai (August), Guangzhou (October), and Chongqing (August). A box model is used to simulate the OH, $HO_2$, and $RO_2$ concentrations. The key processes are elucidated via explicit radical budget analysis. The formation mechanism of secondary pollutants (ozone and particle precursors) is investigated using a chemical model. The major findings of this study are discussed below.

1) The OH reactivity is used to demonstrate the burden of air pollutants. The modeled OH reactivity shows typical diurnal profiles, being maximum in the morning and minimum in the afternoon. The OH reactivity is highest in Guangzhou (20–30 $s^{-1}$), followed by Beijing and Chongqing (15–25 $s^{-1}$); it is smallest in Shanghai (<15 $s^{-1}$). More than 50% of the OH reactivities are contributed by inorganic species, that is, CO and $NO_x$. The measured AVOCs contribute between 14% to 26% to the total OH reactivity. In Guangzhou, their contribution and absolute reactivity (6 $s^{-1}$ compared with 2–3 $s^{-1}$ in other cities) are the highest. The primary OH reactant is propene in all cities.

2) Modeled OH concentrations show a distinct diurnal variation. The mean OH diurnal profile maximum is the largest in Beijing and Shanghai (7 × $10^6$ $cm^{-3}$) and decreases to 4 × $10^6$ $cm^{-3}$ in Chongqing and 2 × $10^6$ $cm^{-3}$ in Guangzhou. The modeled OH concentrations correlate with the photolysis frequencies ($R^2 > 0.7$), with a correlation slope of 2.0 × $10^{11}$ $cm^{-3}$ $s^{-1}$ (3.0 × $10^{11}$ $cm^{-3}$ $s^{-1}$ in Shanghai). The peroxy radical concentrations are the highest in Chongqing, with diurnal maxima of 5 × $10^8$ $cm^{-3}$ for $HO_2$ and 7 × $10^8$ $cm^{-3}$ for $RO_2$ due to the relatively high VOCs/$NO_x$ ratio.

3) The dominant radical sources are photolysis reactions including HONO, $O_3$, HCHO, and other carbonyl compounds. The photolysis of HONO and HCHO accounts for ~50% of the primary sources. The mean diurnal maximum is the largest in Beijing (5 ppbv/h), followed by Shanghai (4.6 ppbv/h) and Chongqing (4.3 ppbv/h); it is the smallest in Guangzhou (3.2 ppbv/h) due to the

later observation period. However, alkene ozonolysis significantly contributes to the radical sources in Guangzhou (43% of the total primary sources under daytime conditions).

4) The daily integrated local ozone production rate is the largest in Beijing (136 ppbv); it is 92, 40, and 105 ppbv in Shanghai, Guangzhou, and Chongqing, respectively. The measurement sites represent city center conditions, where ozone precursors are freshly emitted. With the advection of fresh-emitted air masses, $NO_x$ and VOCs undergo efficient photochemical processes, producing a large amount of ozone, which is transmitted to downwind regions. The outflow of $O_x$ occurs during the afternoon in Beijing, Shanghai, and Chongqing and after the sunset in Guangzhou.

5) The ozone production is VOC-limited in all cities because the $L_N/Q$ ratio is greater than 0.5. In addition, the RIR values of AVOC are the highest in all cities compared with CO, $NO_x$, and isoprene. The speciation shows that alkanes and alkenes are major contributors to the total OH reactivity, except in Guangzhou. With respect to the ozone formation potential, aromatics are dominant species due to their large carbon numbers and high MIR value. Aromatics contribute up to 70% of the share of OFPs in Guangzhou, followed by Shanghai (55%), Beijing (43%), and Chongqing (42%). In comparison, alkanes are the major contributors to the mixing ratios but have limited impact on the ozone formation and radical chemistry. To avoid a bias in the understanding of the photochemistry process, it is more appropriate to use the concept of OH reactivity.

6) The coexistence of high OH and $NO_2$ concentrations results in a fast nitric acid production rate. The partitioning between $HNO_3$ and $NO_3^-$ was analyzed using a thermal dynamic model (ISORROPIA2). In the presence of abundant ammonium, the photochemically produced $HNO_3$ efficiently partitions to an aerosol phase under high-RH conditions.

**Author contribution**

YZ, KL organized the field campaign. KL and YZ designed the experiments. KL, and ZT analyzed the data. ZT wrote the manuscript. All authors contributed to measurements, discussing results, and commenting on the manuscript.

**Acknowledgment**

This work was supported by the National Natural Science Foundation of China (**21522701**, **91544225**, **21190052**, **41375124**), the National Key R&D Plan of China (**2017YFC0210000, 2017YFC0213000**), the National Science and Technology Support Program of China (**2014BAC21B01**), the Strategic Priority Research Program of the Chinese Academy of Sciences (**XDB05010500**), the BMBF project: ID-CLAR (**01DO17036**), Shanghai Science and Technology Commission of Shanghai Municipality (**18QA1403600**), and Shanghai Environmental Protection Bureau (**2017-2**).

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

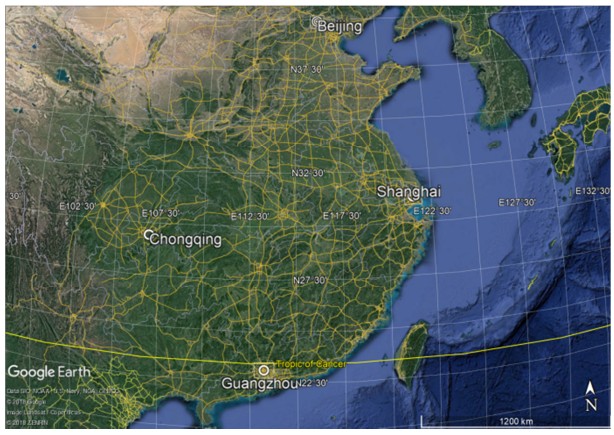

**Figure 1. Locations of the four Chinese megacities.**

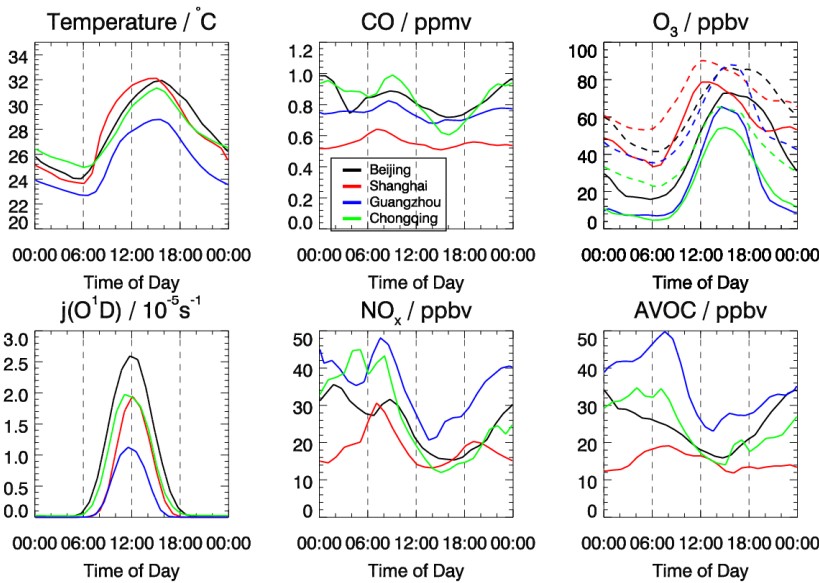

**Figure 2. Mean diurnal variations of the measured temperature, CO, O₃, j(O¹D), NOₓ, and anthropogenic volatile organic compounds (AVOC) based on four field studies. The Oₓ is denoted by dashed lines in the same panel as O₃.**

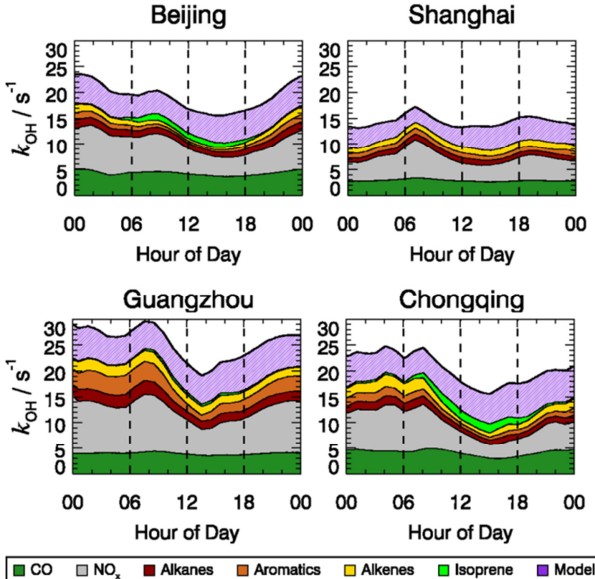

**Figure 3. Mean diurnal profiles of the contributions from all measured species to the OH reactivity in Beijing, Shanghai, Guangzhou, and Chongqing. The filled areas represent different atmospheric constituents. The model denotes the sum of the model-generated species such as formaldehyde and acetaldehyde.**

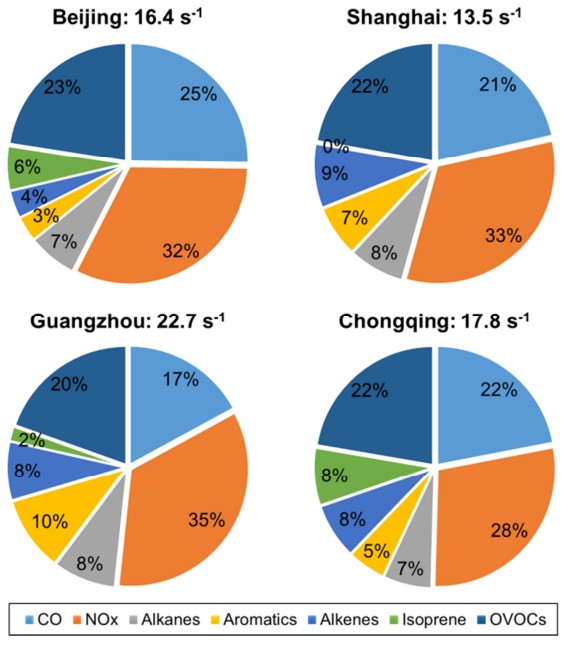

**Figure 4. Contributions of different atmospheric constituents to the OH reactivity in Beijing, Shanghai, Guangzhou, and Chongqing. The OVOC contributions were simulated using a box model.**

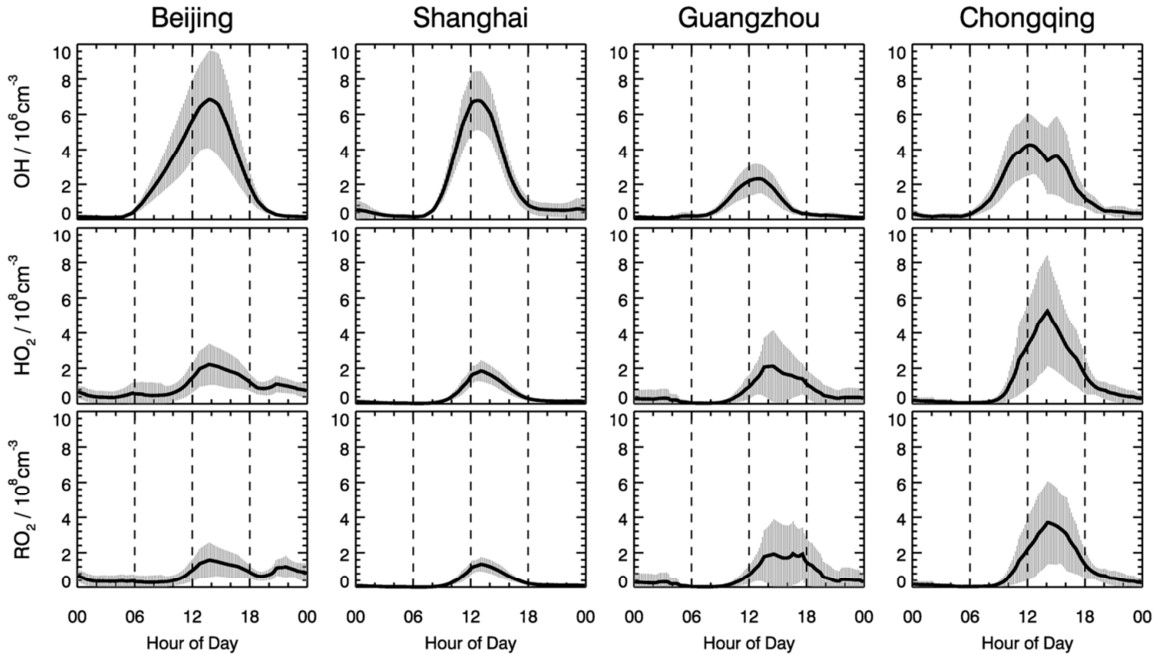

**Figure 5.** Mean diurnal profiles of the OH, HO₂, and RO₂ concentrations modeled for the four measurement sites. The vertical bars denote the daily variability of the model-calculated radical concentrations.

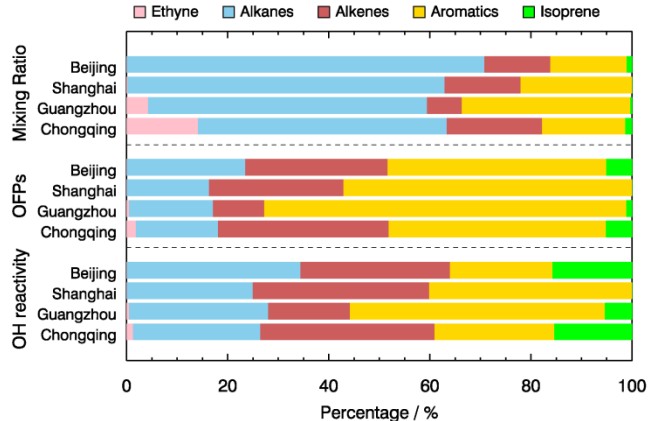

**Figure 6.** Group compositions (mixing ratios) in percentage for primary VOCs and their shares in the OFPs and OH reactivity for four cities.

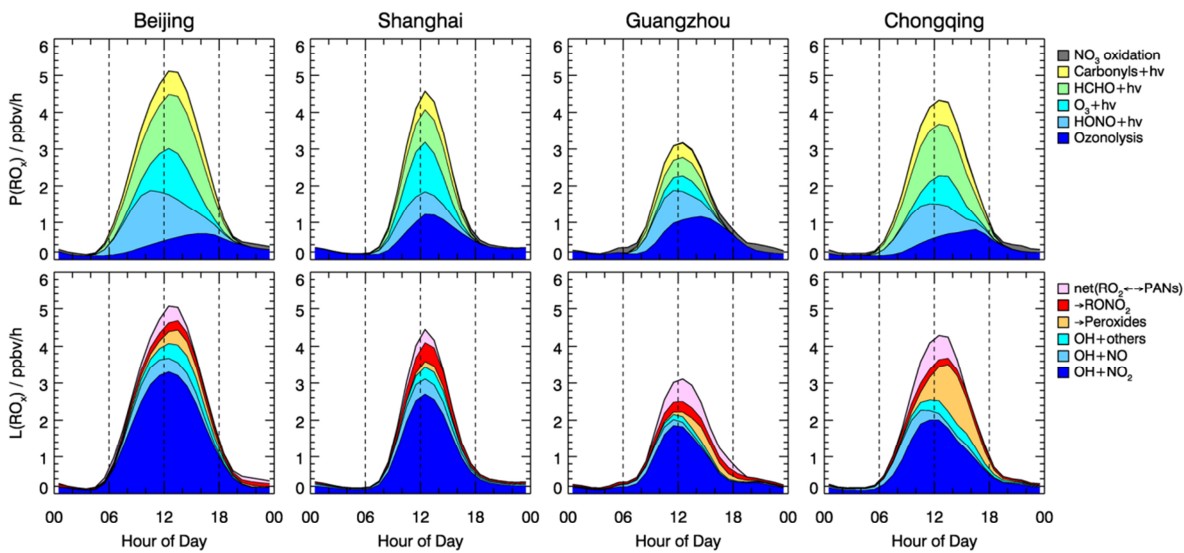

**Figure 7. Hourly averaged primary sources and sinks of ROₓ radicals derived from model calculations for four measurement sites.**

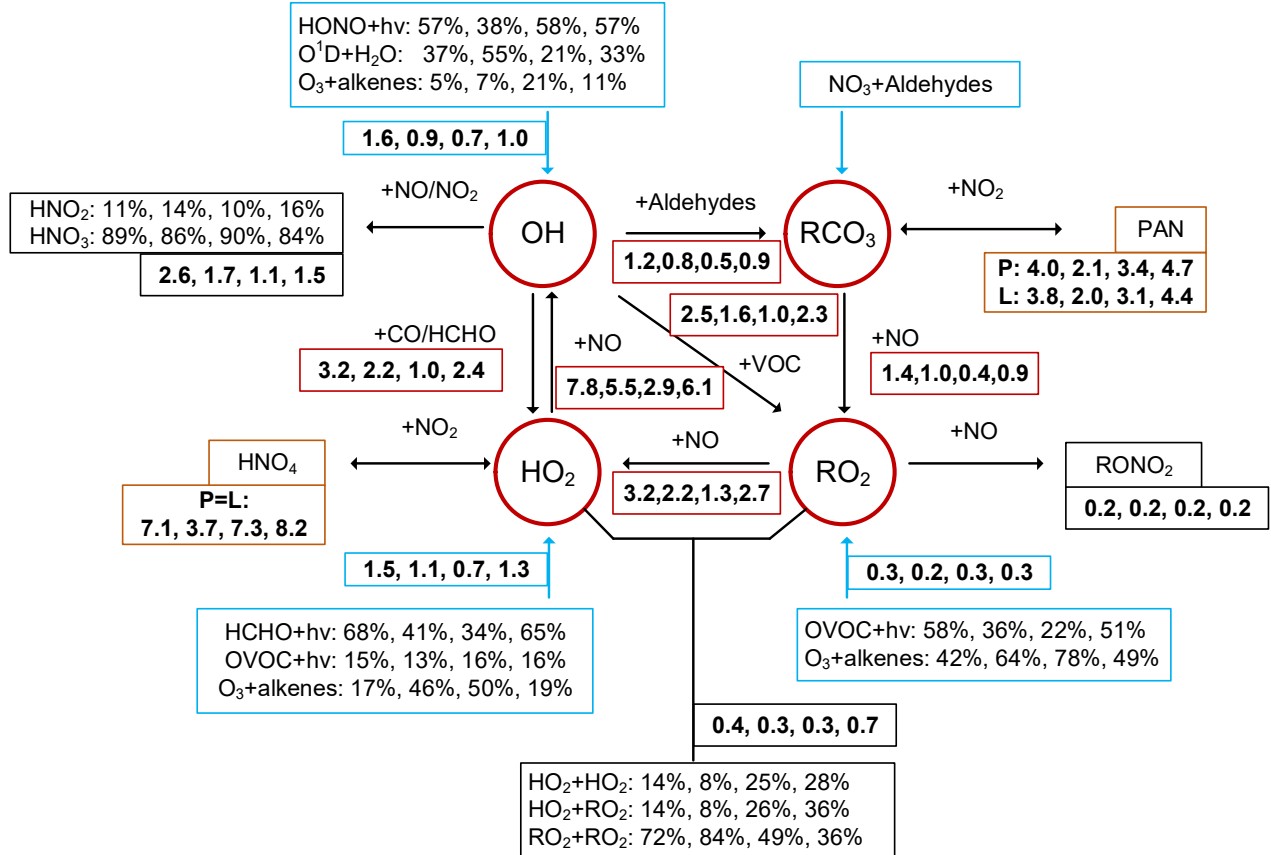

**Figure 8. Comparison of the OH–HO₂–RO₂ radical budget in four cities under daytime conditions (6 am to 6 pm LT). The numbers are sorted from left to right in the order of Beijing, Shanghai, Guangzhou, and**

**Chongqing. The blue, black, red, and yellow boxes denote the** primary radical **sources, radical termination, radical propagation, and equilibrium between radicals and reservoir species, respectively.**

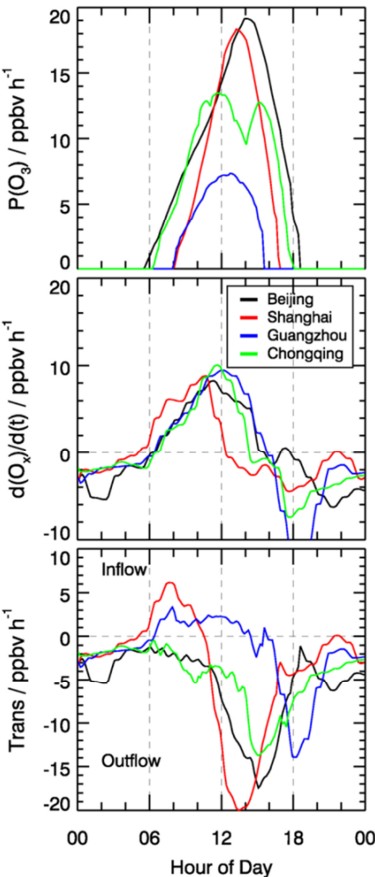

**Figure 9. *In situ* ozone budget analysis in four cities. The upper panel denotes the local ozone production rate P(O₃) derived from the model calculation. The middle panel denotes the derivatives of the observed $O_x$ concentrations $d(O_x)/d(t)$. The bottom panel denotes the difference between P(O₃) and $d(O_x)/d(t)$, which indicates the effect of local chemical production on transportation (see text).**

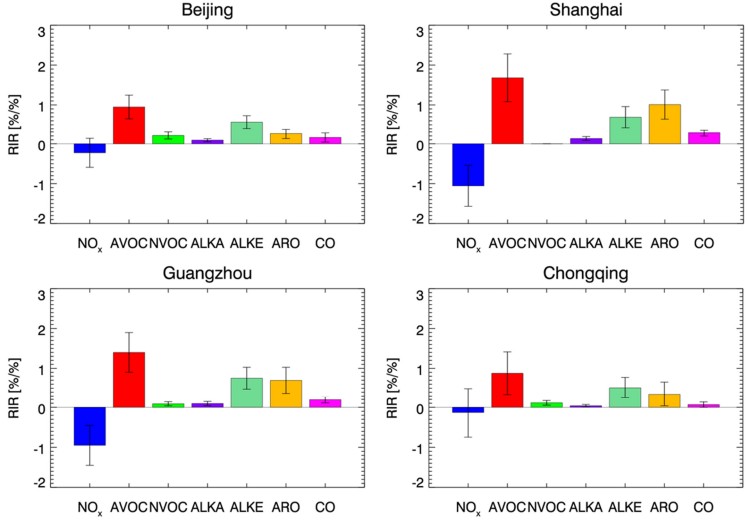

**Figure 10. Relative incremental reactivity analysis for NOx, AVOC (anthropogenic volatile organic compounds), CO, and NVOC (natural volatile organic compounds; in this study only isoprene was considered) at four sites.**

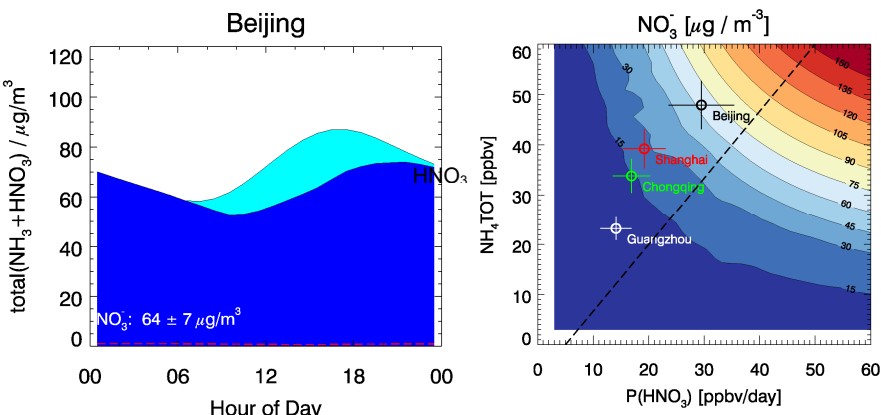

**Figure 11. Modeled nitrate production from gas phase oxidation. (a) The mean diurnal profile of the modeled total nitrate concentration and its gas–particle partitioning. (b) Functional dependence of particulate nitrate concentrations on the daily integrated nitrate production rate and total ammonium concentration.**

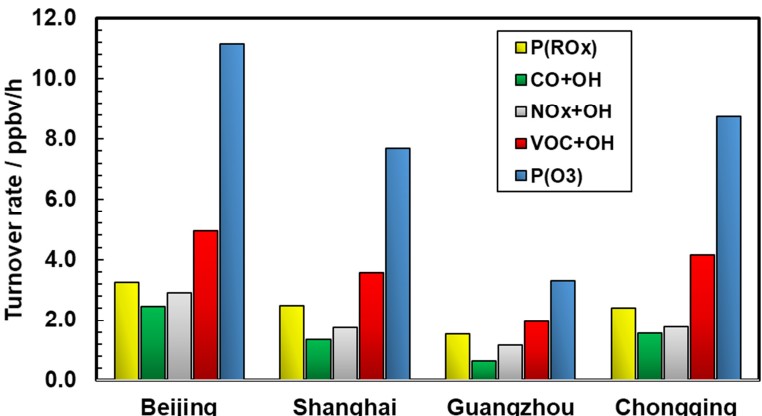

**Figure 12. Intercomparison of the atmospheric oxidation rates for four megacities.**