# Peer review of "Daytime atmospheric oxidation capacity in four Chinese megacities during the photochemically polluted season: A case study based on box model simulation"

_Atmospheric Chemistry and Physics, 2018_

## Referee Comment (RC1) · Anonymous Referee #1 · 9 Dec 2018

The authors present the photochemical oxidation processes simulated by a chemical box model with constraints of field measurement data in four Chinese megacities. The ROx radical budget, OH reactivity, ozone production, and particulate nitrate formation are assessed. The results provide some insights into the formation of secondary pollution such as ozone and nitrate, which are helpful for formulating the air pollution control policy in the polluted regions of China. Hence, this manuscript can be considered for publication after the following comments being properly addressed.

Major Concerns:

A major concern is on the representativeness of measurement results given their relatively short periods, i.e. 1-2 weeks in each city. Are the measured conditions of ozone pollution and ozone precursors typical for the individual cities? Can these results reflect the difference in the photochemical pollution condition and chemical environment among these cities, e.g., less serious pollution in Chongqing? A comment on this would be helpful.

Another concern is on the lack of direct measurements of some key radical precursors, such as HONO and carbonyls. This would influence the modeling results of radical budget as well as ozone and nitrate formation. In this study, the authors scaled the HONO concentrations to $0.02*NO_2$, and performed sensitivity run by turning off the scaling. But the key question here is if the scaling factor of 0.02 is appropriate. What are the scaled HONO levels in the model for the four cities? Using a different scale factor may change the budget of primary $ROx$ sources and OH levels. The authors are recommended to perform more sensitivity tests with different scale factors and comment on its influence on the major conclusions of this study. Moreover, how are the carbonyls treated within the model? What are the measured or simulated levels of major carbonyls such as formaldehyde, etc.?

Section 3.1: a detailed discussion on the concentrations and speciation of VOCs in the four cities are needed. It is much helpful for the readers to understand the different chemical mix in these cities and to better judge the following modeling results. A table summarizing the measured VOC species and related parameters would be good.

Section 3.3.2 and Figures 7-8: it is interesting that ozonolysis reaction of VOCs is identified as a significant daytime source of $ROx$ radicals in four cities. This source is usually considered to be not important at daytime as it only occurs for unsaturated VOCs which are generally at low concentration levels at daytime. So what are the major VOC species contributing to this radical source, and what are their concentration levels in these four cities? In addition, some studies found photolysis of OVOCs is an important radical source. However, its contributions estimated in this study were quite

small (3-6%) in all four cities, and the production rates of RO2 radicals (P(RO2); 0.2-0.3 ppb/h) were also much smaller compared to the other studies. These results are a little bit strange, and the authors may need examine what OVOC species are considered in the model. More discussions of the radical budget analysis are needed.

Specific Comments:

Title: Atmospheric oxidation capacity in four Chinese megacities. . .

Page 1, Line 40: is one of the major threats. . .

P2, L1-2: it would be better to introduce Chongqing here after introducing the other three megacities. . .

P2, L4: have declined. . .characterized by high concentrations of ozone and fine particles. . .

P2, L11: the role of. . .

P3, L7: are located. . .

P3, L16: according to Table 1, the measurement in Beijing was in July, not in June.

P3, L21: Instrumentation

Section 2.2: a table summarizing all of the measurement species (especially the VOC species) and techniques is needed, maybe in the supplement, for better understanding the present study.

P3, L29-30: rephrase this sentence.

P4, L5: AHC is used here but AVOCs is used later. Keep them consistent and spell out them at their first appearance.

P5, L1-8: it seems that the discussion in this paragraph is not relevant here. . .

P6, L4-5, "The maximum of OH. . .": rephrase this sentence.

P6, L13-14, "the larger correlation slope…": rephrase this sentence since solar radiation cannot directly converts to radicals…

P9, L33-34: a brief discussion of the NHC levels would be much helpful here.

P9, L35-36: "reducing NOx could lead to increase in Ox concentrations" is not absolutely correct. There is a threshold below which the NOx emissions were reduced to, the ozone would be significantly decreased.

P10, L6, "In another word, if HONO is": uncomplete sentence.

P10, L34: change decomposition to deposition

Section 4.2: is the nocturnal nitrate formation from the N2O5-related processes taken into account in the present analysis?

P12, L25: change Shang to Shanghai

Figure 2: Anthropogenic Volatile Organic Compounds (AHC): keep consistent.

Figure 3: Chengdu should be Chongqing? In addition, what does "model" mean in the figure legend?

Figure 6: what does "variability" stand for?

---

## Referee Comment (RC2) · Anonymous Referee #2 · 17 Dec 2018

This manuscript describes constrained photochemical modeling of four large urban areas in China. The paper is difficult to read due to organization, presentation and grammar. I found it difficult to understand what exactly was modeled or measured and how. The conclusion that all four cities are VOC limited is probably correct and probably worth noting for these cities. However, some of the conclusions such as the importance of some radical sources is more difficult to justify as they are not based on observations. I have included major and minor comments below. However, please note the manuscript needs significant editing for style and grammar beyond my suggestions.

I think this paper is only publishable after major revisions. 1) The title and abstract indicate that the "atmospheric oxidation capacity" is the focus of the paper. That is fine but this term should be defined instead of vaguely described as in the first line of the abstract. Once defined the values for the different cities should be reported – preferably in the abstract and in the results. I would define the AOC as the reactions of OH, ozone, NO3, etc. that lead to oxidation of an atmospheric component. I would expect units of something like the amount of oxidized molecules per time. The authors only include OH in their reporting of AOC and only vaguely report the values. This needs to be tightened up. I am sure that OH dominates but ozone and NO3 may be important at night and this should needs to be at least mentioned. 2) When I read the abstract, I thought this was going to be more of an observational study than a modeling project. I expected to see observations of OH, HO2, etc. So I recommend stating clearly that this is a photochemical modeling study constrained by observations of NOx, ozone, etc. For example, I initially thought that OH reactivity was measured in this study instead of being calculated from VOC observations. So please make it clear what is measured and how. The lack of any detail in the instrumentation section is unacceptable in my opinion. I suggest that a table be made of every parameter that is measured, including the method, and a reference. I realize standard commercial instruments perform some of the measurements such as ozone and CO. However, many of the measurements are not run of the mill. In particular, there needs to be a reference to the VOC measurement method and a list of measured compounds and detection limits listed in the supporting information. In addition, I do not know what NO2 chemical conversion to NO means as stated on line 25 of page 3. This needs to be described and the probability of interference from PAN needs to be discussed. I would expect at least 5 ppbv of PAN in areas such as Beijing during the day. This could lead so a significant interference in NO2. Please also report in more detail on the VOC observations. I think at least averages of the top 10 or 5 VOC in terms of OH loss should be listed for each city. I like the graphs in the supplement but the scales on many of the graphs don't make sense. Often the parameter graphed only

goes up to 10 or 20% of full scale making it impossible to see what is going on. I don't think keeping consistent axes between different cities is worth not being able to read the graph. 3) The reporting of OH reactivity could be made much more simple as well. Perhaps having a section in the results showing the VOC observations separately would be less confusing. You could then have a following section on the calculated OH reactivity. I really recommend limiting the discussion in these sections and focusing on the results. For example, the paragraph on line 1 page 5 stating that OH reactivity can be measured in 3 ways made me think for some time that this was a measured quantity in this work. 4) I highly recommend being more explicit on what is derived from the model or parameterized. For example, I don't think formaldehyde or acetaldehyde are measured but are model predicted. If so this needs to be described and predicted levels compared to observations if available. This will certainly impact the radical budget as well as the production rate of PAN relative to HNO3. So I suggest a table of model parameters that are predicted, constrained, and parameterized. I also suggest that the model results be presented in an organized manner in the results section. There is a lot of discussion throughout section 3 that should probably be in section 4. 5) The very simple parameterization of HONO as being 2% of NO2 is somewhat troubling. I am surprised that it would be that simple especially as a function of the time of day. I think this assumption needs to be better justified and probably looked at to determine the sensitivity, i.e. some case studies with different assumptions are probably needed. This is also another reason to describe the NO2 measurement in more detail. 6) I am not sure the ISOROPPIA modeling adds much to the paper especially as there are no measurements of ammonia or nitric acid. I certainly realize that if there is a large excess of ammonia that this will drive nitric acid into the aerosol. However, I am not sure the nitric formation rate vs. loss rate to aerosol versus dry and wet deposition can be suitably treated in this work to allow for quantitative predictions of ammonium nitrate aerosol. So I recommend removing from the paper and perhaps replacing with a simple discussion. This discussion could also mention that cutting down NOx may lead to enhanced ozone production but it will cut down on particulate nitrate as well.

---

## Referee Comment (RC3) · Anonymous Referee #3 · 21 Dec 2018

This paper presents results from a 0-D box model constrained by observations of radical sources and sinks in order to evaluate the oxidation capacity of several Chinese megacities, including Beijing, Shanghai, Guangzhou, and Chongqing. The models suggest that while there are similarities in the chemistry of each urban area, such as ozone production being VOC-limited in each, there are some distinct differences in predicted radical concentrations, rates of ozone production, and OH reactivity, which may help provide insights into specific control strategies for each area.

While the paper provides some interesting contrasts between the cities, it is unfortu-

nately somewhat difficult to read due to issues related to both the amount of information and how it is presented as well as style and grammar. In particular, section 3.3.2 describing the radical budget analysis reads more like a stream of thought rather than an organized discussion.

A major assumption in the paper is that the model can accurately reproduce concentrations of OH, HO2, and RO2 radicals in order to predict the oxidation capacity of each region. Unfortunately, there is no discussion of whether this is a reasonable assumption. As mentioned in the introduction, previous measurements of radical concentrations in urban areas often exhibit significant discrepancies with model predictions, suggesting that chemical models are unable to accurately reproduce the oxidation capacity of these areas (see for example Whalley et al. (2018), Griffith et al. (2016), in addition to references cited in the Introduction). As summarized in the Lu et al. (2018) review cited in the paper, ". . .current tropospheric chemical mechanisms cannot explain the OH radical concentrations in China, which strongly underestimated the OH concentrations and the local ozone production for the low and high NOx range, respectively." The authors should expand the discussion of these discrepancies and discuss in much more detail their potential impact on their model predictions and conclusions.

Additional comments:

1) The paper would benefit from a more detailed description of what was measured and how they were measured, perhaps with a table in the supplement. In particular, the specific VOCs that were measured should be described in more detail.

2) Instead of just showing the total AHC (or preferably AVOC as indicated elsewhere in the manuscript), it would be more informative to illustrate the diurnal mixing ratios of some important individual VOCs that demonstrate the similarities and differences in the areas as described in the manuscript.

3) In addition, it should be clarified which VOCs and/or OVOCs were measured and which were modeled as part of the radical budget. For example, were HCHO and other

carbonyls measured or was their contribution to radical production based on modeled concentrations?

---

## Author Response (AR1)

*The authors present the photochemical oxidation processes simulated by a chemical box model with constraints of field measurement data in four Chinese megacities. The ROx radical budget, OH reactivity, ozone production, and particulate nitrate formation are assessed. The results provide some insights into the formation of secondary pollution such as ozone and nitrate, which are helpful for formulating the air pollution control policy in the polluted regions of China. Hence, this manuscript can be considered for publication after the following comments being properly addressed.*

**Answer:**

We thank the comments and suggestions from the reviewers, which help to improve the manuscript considerably. The response and changes are listed below.

**Major Concerns:**

*A major concern is on the representativeness of measurement results given their relatively short periods, i.e. 1-2 weeks in each city. Are the measured conditions of ozone pollution and ozone precursors typical for the individual cities? Can these results reflect the difference in the photochemical pollution condition and chemical environment among these cities, e.g., less serious pollution in Chongqing? A comment on this would be helpful.*

**Answer:**

We compared the measurement presented in this study to the observation obtained by the environmental monitor stations. The Chinese EPA station data of the same year are derived. We added figures of the EPA station results in four cities in the supplement (Fig. S5). In general, the $O_3$ concentrations presented in this study are comparable to the maximum of monthly averaged O3 concentrations derived from the EPA monitor station data for the same year. As shown in Fig. S5, the $O_3$ concentrations are in general lower in Chongqing than those observed in the other three cities.

[Figure]

**Figure S5. The monthly averaged diurnal profiles of measured O$_x$, O$_3$, NO$_2$, CO, PM2.5 concentrations in (a) Beijing, (b) Shanghai, (c) Guangzhou, and (d) Chongqing.**

We added sentences in Line 12 Page 5 "Given the relatively short periods for these campaigns, one concern is about the representativeness of measurements. We compared the observation from these

intensive campaigns to the routine measurement obtained in the environmental monitor stations operated by the Chinese environmental protection agency (Fig. S5). We found that the mean diurnal profiles of $O_3$ and $O_x$ obtained in all sites are comparable to the highest monthly averaged diurnal profiles for the same city (bias < 20%). The relatively small $O_3$ and $O_x$ concentrations observed in Chongqing compared to other cities (Fig. 2) is consistent with the environmental monitor stations observation (Fig. S5). Therefore, it suggests that the ozone pollution is less severe in Chongqing compared to the megacities in eastern China."

*Another concern is on the lack of direct measurements of some key radical precursors, such as HONO and carbonyls. This would influence the modeling results of radical budget as well as ozone and nitrate formation. In this study, the authors scaled the HONO concentrations to 0.02\*NO2, and performed sensitivity run by turning off the scaling. But the key question here is if the scaling factor of 0.02 is appropriate. What are the scaled HONO levels in the model for the four cities? Using a different scale factor may change the budget of primary ROx sources and OH levels. The authors are recommended to perform more sensitivity tests with different scale factors and comment on its influence on the major conclusions of this study.*

**Answer:**
Although the HONO to $NO_2$ ratio is relative robust and constant as reported in other field campaigns (Elshorbany et al., 2012), such simple parameterization could increase the uncertainty of our model calculation. We performed more sensitivity tests to demonstrate the uncertainty for this simple parameterization and found the model results are not so sensitive to the parameterization (Fig. S8). We added a discussion in Line 9 Page 5 "Although the HONO to $NO_2$ ratio is relative robust and constant as reported in other field campaigns (Elshorbany et al., 2012), such simple parameterization could increase the uncertainty of our model calculation. To further investigate the uncertainty from this simple parameterization, the scaling factor is varied from 0.015 to 0.03. The modelled OH concentrations change by less than 10 % if the HONO scaling factors change by 50% (Fig. S8). Besides, the modelled $HO_2$ and $RO_2$ concentrations are relatively stable with different HONO scaling factors. The different scaling factors also have impact on the model generated species, e.g. HCHO (Table S4). In fact, the higher HONO concentrations lead to more active photochemical reactions and more HCHO production. The higher HCHO concentrations could further enhance the photochemistry by more radical photolytic sources in return. Therefore, the higher (lower) modelled radical concentrations due to increase (reduce) the HONO scaling factors are also affected by the corresponding change in modelled HCHO concentrations. This demonstrates the nonlinearity of the photochemical system. Nevertheless, the parameterized HONO

concentrations are in the range of 0.3 to 0.6 ppbv during daytime (Table S4), which are consistent with previous in-situ measurements in urban areas (Lu et al., 2013;Li et al., 2010;Ren et al., 2003;Kanaya et al., 2007). To evaluate the impact of missing HONO source on the radical chemistry, we switched off the scaling between HONO and $NO_x$ in a sensitivity test (Fig. S8). Therefore, the results show that OH concentrations reduce by about 20% if the only homogenous source is considered. The modeled $HO_2$ and $RO_2$ concentrations are also reduced by 15-20% (Table S4)."

[Figure]

**Figure S8. Mean diurnal profiles of modeled OH, HO2, RO2 concentrations and kOH in four measurement sites. Black: model base case (HONO=0.02\*NO₂); Red: model sensitivity test M1 (HONO=0.03\*NO₂); Blue: model sensitivity test M2 (HONO=0.015\*NO₂); Green: model sensitivity test M3 (HONO unscaled but simulated free by the box model).**

*Moreover, how are the carbonyls treated within the model? What are the measured or simulated levels of major carbonyls such as formaldehyde, etc.?*

**Answer:**

We added the discussion on the OVOC in Line 34 Page 8 "The OVOCs concentrations are simulated by the box model. The modelled HCHO concentrations are in the range of 3 to 8 ppbv (Fig. S7), which are consistent with the previous studies in these regions (Zhang et al., 2012;Song et al., 2018;Chen et al., 2016;Tang et al., 2009). The modelled acetaldehyde concentrations are in the range of 2 to 3 ppbv in Beijing, Shanghai, and Chongqing but 1 ppbv larger in Guangzhou because the larger contribution of aromatics VOCs which produce acetaldehyde from their OH degradation."

[Figure]

**Figure S7. The mean diurnal profiles of modelled formaldehyde (HCHO), acetaldehyde (ACD) and peroxyacetyl nitrate (PAN) concentrations in four cities.**

*Section 3.1: a detailed discussion on the concentrations and speciation of VOCs in the four cities are needed. It is much helpful for the readers to understand the different chemical mix in these cities and to better judge the following modeling results. A table summarizing the measured VOC species and related parameters would be good.*

**Answer:**

We added a table showing top 10 $k_{OH}$ contributing VOCs in Table 2 and a table showing all measured VOC and their concentrations in supplement (Table S3). The mean diurnal profiles of top 10 VOC are added in supplement (Figure S6).

**Table S3 Summary of measured and modelled species**

| Species | Parameters |
|---|---|
| Measured | T, P, RH, photolysis rate, NO, NO$_2$, O$_3$, C2-C8 alkanes, C2-C6 alkenes, C6-C10 aromatics |
| Modelled | OH, HO$_2$, RO$_2$, $k_{OH}$, OVOCs (including formaldehyde, acetaldehyde, methacrolein, other aldydes, glyoxal , acetones, methyl vinyl ketone, other ketones, methanol, ethanol, phenol, formic acid, acetic acid and higher acids, and so on) |
| Scaled | HONO (=0.02×NO$_2$) |

[Figure]

**Figure S6. The mean diurnal profiles of top 10 $k_{OH}$ contributing VOCs concentrations in Beijing, Shanghai, Guangzhou, and Chongqing.**

We move the discussion of VOC speciation to section 4.1. We added a discussion on the concentrations and speciation of VOCs in the four cities in the end of the section "The top 10 OH reactivity contributing VOCs are summarized in Table 2. The order of VOCs is sorted by the averaged OH reactivity for four cities. Among all, propene are the most important OH reactivity contributor, which contributed about

0.4~0.6 s$^{-1}$ (Table 2). The small VOCs (propene, ethane, ethene) are relatively important with respect to OH reactivity. 9 out of the top 10 VOCs are alkenes and aromatics (except ethane). In Guangzhou, the xylene (m,p-, and o-) and toluene are also important OH reactants, consistent with the inventory study (Zheng et al., 2009). The diurnal profiles are shown in Fig. S6. The observed anthropogenic VOC concentrations show typical diurnal profile that increase during night and decrease during afternoon. One exception is Shanghai site, the mean diurnal profiles of propene and 1,2,4-trimethybenzene are flat, while that of styrene shows peak around noontime, indicating unique VOC emission feature in that site.

The OVOCs concentrations are simulated by the box model. The modelled HCHO concentrations are in the range of 3 to 8 ppbv (Fig. S7), which are consistent with the previous studies in these regions (Zhang et al., 2012;Song et al., 2018;Chen et al., 2016;Tang et al., 2009). The modelled acetaldehyde concentrations are in the range of 2 to 3 ppbv in Beijing, Shanghai, and Chongqing but 1 ppbv larger in Guangzhou because the larger contribution of aromatics VOCs which produce acetaldehyde from their OH degradation. ".

*Section 3.3.2 and Figures 7-8: it is interesting that ozonolysis reaction of VOCs is identified as a significant daytime source of ROx radicals in four cities. This source is usually considered to be not important at daytime as it only occurs for unsaturated VOCs which are generally at low concentration levels at daytime. So what are the major VOC species contributing to this radical source, and what are their concentration levels in these four cities? In addition, some studies found **photolysis of OVOCs** is an important radical source. However, its contributions estimated in this study were quite small (3-6%) in all four cities, and the production rates of RO2 radicals (P(RO2); 0.2-0.3 ppb/h) were also much smaller compared to the other studies. These results are a little bit strange, and the authors may need examine what OVOC species are considered in the model. More discussions of the radical budget analysis are needed.*

**Answer:**

As shown in the newly added Table 2, 5 out of the top 10 VOC could react with O$_3$ to produce ROx radicals. However, the reaction rates with O$_3$ and with OH are different for different alkenes. We added a discussion in Line 17 Page 9 "The ozonolysis reactions mainly contributed by trans-2-butene in Beijing (55%), Guangzhou (42%), and Chongqing (39%), whose concentrations are in the range between 0.1 to 0.3 ppbv (Fig. S6). Although trans-2-butene is only the 8th important VOCs with respect to OH reaction (Table 2), it become the most important O$_3$ reactants producing RO$_x$ radicals due to its fast reaction rate with O$_3$ (1.9×10$^{-16}$ cm$^{-3}$s$^{-1}$ compared to 1.0×10$^{-17}$ cm$^{-3}$s$^{-1}$ of propene, rate constant derived from MCM3.3.1 (http://mcm.leeds.ac.uk/MCMv3.3.1/home.htt)). In Shanghai, propene becomes the most important alkene

with respect to $O_3$ reaction, which accounts for about 42% of the total alkene ozonolysis reactions. Actually, the relatively high contribution from alkene ozonolysis to the $RO_x$ radical primary sources could be one of the important characteristics for $RO_x$ radical primary sources in Chinese megacity. The importance of alkene ozonolysis was also found in Santiago, Chile (Elshorbany et al., 2009) and Essex, UK (Emmerson et al., 2007), where alkene ozonolysis contributed about 20% to the total radical primary production."

The role of OVOCs (except HCHO) photolysis are highly variable from literatures, which demonstrates the fate OVOCs highly depends on chemical conditions. We added a discussion on the role of OVOC photolysis to radical chemistry from Line 30 Page 10 to Line 2 Page 11 to replace the sentences "The $RO_2$ primary source strength is in the range of 0.2 to 0.3 ppbv/h, which is mainly contributed by alkene ozonolysis and OVOC photolysis (Fig. 8). In this study, the OVOC photolysis mainly includes carbonyl-containing compounds (e.g. acetaldehyde, aldehydes with carbon numbers larger than 3), which are generated by the box model. The modelled acetaldehyde concentrations are in the range of 2 to 4 ppbv (Fig. S7), consistent to the observations in Beijing (Chen et al., 2016) and Hong Kong (Lyu et al., 2016). The photolysis rate of carbonyl-containing compounds (except HCHO) is about one third to a quarter of the HCHO photolysis rate. In comparison, this ratio could be close to or even higher than 1 in other urban studies (Ren et al., 2013;Volkamer et al., 2010;Emmerson et al., 2007;Michoud et al., 2012;Whalley et al., 2018;Xue et al., 2016). In contrast, the relatively small contribution from other carbonyl-containing compounds photolysis than HCHO photolysis were reported in an urban and suburban site in Hong Kong (Lyu et al., 2016), where the acetaldehyde concentrations were about 1 to 2 ppbv, comparable to our model simulation. Such large discrepancy in the role of other OVOC photolysis to the radical production highlights the importance to measure these radical precursors in the future studies."

**Specific Comments:**

*Title: Atmospheric oxidation capacity in four Chinese megacities*

**Answer:**

We changed the title to be "Daytime atmospheric oxidation capacity in four Chinese megacities during photochemical polluted season: a case study based on box model simulation".

*Page 1, Line 40: is one of the major threats*

**Answer:**

Corrected.

*P2, L1-2: it would be better to introduce Chongqing here after introducing the other three megacities*

**Answer:**

We moved the introduction of Chongqing to Line 6 Page 2 "The Chengdu-Chongqing city group (population 90 million) locates in Sichuan Basin (SCB), southwest of China, representing the developing city clusters. Chongqing is the biggest city in the southwest of China, which suffers from severe air pollution as well."

We moved the sentence "Although the new city clusters also suffer from air pollution, only sparse researches have been conducted in these regions, especially for the secondary pollution formation. For the SCB region, only limited studies have been performed regarding the oxidation capacity. Chengdu is evaluated using an observational-based model which found similar radical concentration and ozone production rate (Tan et al., 2018c). The VOC and ozone formation is evaluated in Chongqing (Su et al., 2018;Li et al., 2018b)." to Line 37 Page 2.

*P2, L4: have declined characterized by high concentrations of ozone and fine particles*

**Answer:**

We changed the sentence to be "As a result, the primary pollutant concentrations have declined since then. However, secondary pollution characterized by high concentrations of ozone and fine particle has become the major contributor to air pollution."

*P2, L11: the role of*

**Answer:**

Corrected.

*P3, L7: are located*

**Answer:**

Corrected.

*P3, L16: according to Table 1, the measurement in Beijing was in July, not in June.*

**Answer:**

Corrected.

*P3, L21: Instrumentation*

**Answer:**

Corrected.

*Section 2.2: a table summarizing all of the measurement species (especially the VOC species) and techniques is needed, maybe in the supplement, for better understanding the present study.*

**Answer:**

We added a table to describe the instrumentation in the supplement (Table S1). The measured VOCs listed are presented in Table S3.

*P3, L29-30: rephrase this sentence.*

**Answer:**

We changed the sentence to be "A box model based on the Regional Atmospheric Chemical Mechanism version 2 (Goliff et al., 2013) is used to simulate the concentrations of the OH, $HO_2$ and $RO_2$ radicals concentrations and other unmeasured secondary species concentrations."

*P4, L5: AHC is used here but AVOCs is used later. Keep them consistent and spell out them at their first appearance.*

**Answer:**

We changed all AHC to AVOC. For consistency, we changed NHC to NVOC.

*P5, L1-8: it seems that the discussion in this paragraph is not relevant here*

**Answer:**

We rearrange the paragraph from Line 34 Page 5 to Line 10 Page 6.

P6, L4-5, "The maximum of OH": rephrase this sentence.

**Answer:**

We changed the sentence to be "The peak of OH mean diurnal profile is highest in Beijing and Shanghai $(7 \times 10^6 \ cm^{-3})$."

*P6, L13-14, "the larger correlation slope": rephrase this sentence since solar radiation cannot directly converts to radicals*

**Answer:**

This sentence is only a qualitative description and not related to the content. Hence, we removed this sentence.

*P9, L33-34: a brief discussion of the NHC levels would be much helpful here.*

**Answer:**

We added a sentence in Line 15 Page 13 "The only component of NVOC is isoprene, which are on average 0.3 ppbv in Beijing, 0.1 ppbv in Guangzhou, and 0.4 ppbv in Chongqing, but reduce to negligible in Shanghai (below detection limit, Table S3)."

*P9, L35-36: "reducing NOx could lead to increase in Ox concentrations" is not absolutely correct. There is a threshold below which the NOx emissions were reduced to, the ozone would be significantly decreased.*

**Answer:**

We changed the sentence to be "For $NO_x$, the RIR values are negative indicating the ozone production is in $NO_x$-titration regime. A slight reduction of $NO_x$ could lead to increase in $O_x$ concentrations within the $NO_x$-titration regime."

*P10, L6, "In another word, if HONO is": uncomplete sentence.*

**Answer:**

The sentence should be "The reduction of radical termination by $OH+NO_2$ reaction compensates the reduction of radical primary source due to HONO photolysis. Therefore, a larger negative effect shows up in the RIR analysis in the sensitivity test (Fig. S6)."

*P10, L34: change decomposition to deposition*

**Answer:**

Changed.

*Section 4.2: is the nocturnal nitrate formation from the N2O5-related processes taken into account in the present analysis?*

**Answer:**

Only daytime processes is considered in this study. I added a sentence to state it in Line 1 Page 15 "One should note that the nitration formation from $N_2O_5$ hydrolysis is not taken into account in this study, which could lead to negative bias in the nitrate production."

*P12, L25: change Shang to Shanghai*

*Answer:*

*Corrected.*

*Figure 2: Anthropogenic Volatile Organic Compounds (AHC): keep consistent.*

*Answer:*

*We changed all AHC to AVOC.*

*Figure 3: Chengdu should be Chongqing? In addition, what does "model" mean in the figure legend?*

*Answer:*

*We changed the caption to be "Figure 3. Mean diurnal profile of contributions from all measured species for OH reactivity in Beijing, Shanghai, Guangzhou, and Chongqing. The filled areas represent different atmospheric constituents. The model denotes the sum of model generated species such as formaldehyde, acetaldehyde."*

**Figure 6: what does "variability" stand for?**

*Answer:*

*We changed the caption to be "The vertical bars denote the daily variability of model calculated radical concentrations."*

**Anonymous Referee #2**

*This manuscript describes constrained photochemical modeling of four large urban areas in China. The paper is difficult to read due to organization, presentation and grammar. I found it difficult to understand what exactly was modeled or measured and how. The conclusion that all four cities are VOC limited is probably correct and probably worth noting for these cities. However, some of the conclusions such as the importance of some radical sources is more difficult to justify as they are not based on observations. I have included major and minor comments below. However, please note the manuscript needs significant editing for style and grammar beyond my suggestions. I think this paper is only publishable after major revisions.*

**Answer:**

We thank the comments and suggestions from the reviewers, which help to improve the manuscript considerably. The response and changes are listed below. We also changed the style and grammar of the paper and please find them in the revised manuscript.

*1) The title and abstract indicate that the "atmospheric oxidation capacity" is the focus of the paper. That is fine but this term should be defined instead of vaguely described as in the first line of the abstract. Once defined the values for the different cities should be reported – preferably in the abstract and in the results. I would define the AOC as the reactions of OH, ozone, NO3, etc. that lead to oxidation of an atmospheric component. I would expect units of something like the amount of oxidized molecules per time. The authors only include OH in their reporting of AOC and only vaguely report the values. This needs to be tightened up. I am sure that OH dominates but ozone and NO3 may be important at night and this should needs to be at least mentioned.*

**Answer:**

In the revised manuscript, we restrict the oxidation capacity to daytime photochemical reaction. Also, we point out the focus of this study will be mainly on OH radical chemistry. We changed the title to be "Daytime atmospheric oxidation capacity in four Chinese megacities during photochemical polluted season: a case study based on box model simulation".

In the beginning of the abstract, we changed the Line 25-26 Page 1 "Atmospheric oxidation capacity AOC is the core of converting freshly-emitted substances to secondary products, which are dominated by reaction with hydroxyl radicals (OH) during daytime." We added a sentence to define AOC and restrict it to OH oxidation only in Line 9 Page 3 "AOC can be defined as the sum of respective oxidation rates of trace gases (VOCs, CO, and $NO_x$) by the oxidants (OH, $O_3$, and $NO_3$) (Geyer et al., 2001). Given the relatively importance of OH oxidation during daytime, the AOC is restricted to OH oxidation in this study."

*2) When I read the abstract, I thought this was going to be more of an observational study than a modeling project. I expected to see observations of OH, HO2, etc. So I recommend stating clearly that this is a photochemical modeling study constrained by observations of NOx, ozone, etc. For example, I initially thought that OH reactivity was measured in this study instead of being calculated from VOC observations. So please make it clear what is measured and how. The lack of any detail in the instrumentation section is unacceptable in my opinion. I suggest that a table be made of every parameter that is measured, including the method, and a reference. I realize standard commercial instruments perform some of the measurements such as ozone and CO. However, many of the measurements are not run of the mill. In particular, there needs to be a reference to the VOC measurement method and a list of measured compounds and detection limits listed in the supporting information.*

**Answer:**

As mention previously, we changed the title to be more explicitly demonstrating this is a model-based study. We added a table to describe the instrumentation in the supplement (Table S1). We added a sentence in Line 8 Page 4 "The performance of different instruments is summarized in Table S1."

**Table S1 measured species for ozone pollution analysis and instrument time resolution, accuracy and limit of detection**

| Species | Method | Time resolution | Accuracy (1$\sigma$) | Limit of Detection / ppbv |
|---|---|---|---|---|
| Photolysis frequencies | Actinic flux spectroradiometry | 20 s | ±10 % | Five orders of magnitude lower than maximum at noon |
| $O_3$ | UV absorption | 1 min | 5% | 0.5 |
| NO | Chemiluminescence | 1 min | ±20 % | 60 pptv |
| $NO_2$ | Chemiluminescence | 1 min | ±20 % | 0.3 |
| CO | IR absorption | 1 min | 5% | 4 |
| VOCs | Gas chromatography and mass spectroscopy /flame ionization detector | 1 h | 10%~20% | 0.01~0.2 |

We also prepared a table in supplement to state what are measured, modelled, and parameterized in this study (Table S2). The measured VOCs and their concentrations are presented in table S3. We added a sentence in Line 23 Page 4 "The measured, modelled and parameterized parameters are summarized in Table S3."

We added a description about the VOC measurement in Line 9 Page 4 "VOC measurements (including 55 organic species) were performed by commercial an instrumentation using gas chromatograph (GC) equipped with a mass spectrometer (MS) and a flame ionization detector (FID). In principle, the air sample was drawn into two parallel channels for enrichment by cooling before analysis (Wang et al.,

2014). The VOCs measurements include C$_2$–C$_{11}$ alkanes, C$_2$–C$_6$ alkenes, and C$_6$–C$_{10}$ Aromatics (Table S2)."

**Table S2 Summary of measured VOCs concentration for four campaigns**

| VOC / ppbv | Beijing | | | Shanghai | | | Guangzhou | | | Chongqing | | |
|---|---|---|---|---|---|---|---|---|---|---|---|---|
| | Mean | Median | Max | Mean | Median | Max | Mean | Median | Max | Mean | Median | Max |
| 1,2,3-TRIMETHYLBENZENE | 0.026 | 0.022 | 0.100 | 0.130 | 0.120 | 0.340 | 0.090 | 0.065 | 0.473 | 0.068 | 0.053 | 0.189 |
| 1,2,4-TRIMETHYLBENZENE | 0.098 | 0.085 | 0.370 | 0.160 | 0.150 | 0.610 | 0.199 | 0.121 | 1.140 | 0.225 | 0.161 | 0.756 |
| 1,3,5-TRIMETHYLBENZENE | 0.022 | 0.018 | 0.111 | 0.004 | 0.000 | 0.230 | 0.077 | 0.053 | 0.347 | 0.079 | 0.058 | 0.337 |
| 1-BUTENE | 0.167 | 0.140 | 0.803 | 0.072 | 0.060 | 0.300 | 0.239 | 0.218 | 0.607 | 0.193 | 0.146 | 0.939 |
| 1-HEXENE | Nan | Nan | Nan | 0.323 | 0.280 | 1.870 | 0.074 | 0.048 | 0.429 | 0.065 | 0.062 | 0.193 |
| 1-PENTENE | 0.025 | 0.019 | 0.156 | 0.049 | 0.020 | 0.330 | 0.049 | 0.033 | 0.295 | 0.057 | 0.042 | 0.471 |
| 2,2,4-TRIMETHYLPENTANE | 0.051 | 0.045 | 0.240 | 0.155 | 0.130 | 0.990 | 0.072 | 0.039 | 0.736 | 0.056 | 0.045 | 0.177 |
| 2,2-DIMETHYLBUTANE | 0.020 | 0.019 | 0.070 | 0.149 | 0.140 | 0.410 | 0.099 | 0.063 | 0.898 | 0.054 | 0.037 | 1.236 |
| 2,3,4-TRIMETHYLPENTANE | 0.021 | 0.019 | 0.094 | 0.023 | 0.000 | 0.350 | 0.045 | 0.028 | 0.336 | 0.026 | 0.022 | 0.071 |
| 2,3-DIMETHYLBUTANE | 0.033 | 0.028 | 0.137 | 0.071 | 0.070 | 0.180 | 0.137 | 0.072 | 1.584 | 0.077 | 0.058 | 0.769 |
| 2,3-DIMETHYLPENTANE | 0.049 | 0.038 | 0.469 | 0.027 | 0.000 | 0.490 | 0.111 | 0.061 | 0.667 | 0.056 | 0.040 | 0.353 |
| 2,4-DIMETHYLPENTANE | 0.039 | 0.037 | 0.099 | 0.112 | 0.100 | 0.350 | 0.070 | 0.046 | 0.379 | 0.030 | 0.024 | 0.142 |
| 2-METHYLHEPTANE | 0.016 | 0.014 | 0.050 | 0.002 | 0.000 | 0.210 | 0.066 | 0.046 | 0.440 | 0.039 | 0.032 | 0.162 |
| 2-METHYLHEXANE | 0.061 | 0.055 | 0.227 | 0.000 | 0.000 | 0.000 | 0.273 | 0.174 | 1.391 | 0.133 | 0.095 | 0.976 |
| 2-METHYLPENTANE | 0.226 | 0.206 | 0.983 | 0.265 | 0.230 | 1.610 | 1.066 | 0.557 | 8.730 | 0.360 | 0.268 | 3.827 |
| 3-METHYLHEPTANE | 0.021 | 0.019 | 0.066 | 0.095 | 0.100 | 0.210 | 0.054 | 0.037 | 0.345 | 0.024 | 0.020 | 0.113 |
| 3-METHYLHEXANE | 0.107 | 0.093 | 0.307 | 0.116 | 0.110 | 0.260 | 0.299 | 0.177 | 1.936 | 0.150 | 0.101 | 1.196 |
| 3-METHYLPENTANE | 0.277 | 0.252 | 1.027 | 0.130 | 0.110 | 0.580 | 0.716 | 0.378 | 4.242 | 0.363 | 0.259 | 4.246 |
| BENZENE | 0.909 | 0.780 | 7.830 | 0.413 | 0.350 | 1.240 | 0.989 | 0.560 | 11.448 | 1.080 | 0.995 | 3.749 |
| CIS-2-PENTENE | 0.005 | 0.004 | 0.045 | 0.015 | 0.000 | 0.670 | 0.014 | 0.007 | 0.107 | 0.023 | 0.005 | 0.287 |
| CIS-BUTENE | 0.035 | 0.019 | 0.301 | 0.003 | 0.000 | 0.280 | 0.122 | 0.124 | 0.259 | 0.143 | 0.100 | 1.333 |
| CYCLOHEXANE | 0.079 | 0.058 | 1.048 | 0.097 | 0.080 | 0.320 | 0.222 | 0.103 | 2.180 | 0.079 | 0.064 | 0.293 |
| CYCLOPENTANE | 0.125 | 0.117 | 0.355 | 0.048 | 0.050 | 0.150 | 0.117 | 0.108 | 0.313 | 0.167 | 0.135 | 0.716 |
| ETHANE | 4.896 | 4.570 | 13.941 | 2.432 | 2.300 | 7.570 | 1.952 | 1.661 | 5.029 | 5.145 | 4.957 | 11.305 |
| ETHENE | 2.210 | 2.087 | 7.887 | 0.921 | 0.700 | 5.290 | 1.522 | 1.242 | 6.875 | 4.039 | 3.435 | 11.949 |
| ETHYLBENZENE | 0.335 | 0.257 | 1.636 | 0.355 | 0.290 | 1.460 | 1.322 | 0.782 | 16.959 | 0.625 | 0.480 | 2.176 |
| ETHYNE | Nan | Nan | Nan | 0.025 | 0.020 | 0.130 | 1.355 | 1.263 | 2.949 | 4.123 | 3.649 | 11.352 |
| ISO-BUTANE | 1.836 | 1.747 | 6.574 | 0.779 | 0.650 | 3.760 | 1.884 | 1.536 | 6.630 | 0.652 | 0.542 | 3.581 |
| ISO-PENTANE | 1.414 | 1.326 | 3.941 | 0.691 | 0.560 | 3.110 | 1.205 | 1.079 | 5.581 | 1.987 | 1.412 | 34.131 |
| ISO-PROPYLBENZENE | 0.011 | 0.010 | 0.056 | 0.033 | 0.000 | 0.940 | 0.047 | 0.037 | 0.230 | 0.032 | 0.026 | 0.096 |
| ISOPRENE | 0.272 | 0.208 | 1.289 | 0.000 | 0.000 | 0.110 | 0.126 | 0.088 | 0.809 | 0.404 | 0.332 | 1.641 |
| M-DIETHYLBENZENE | Nan | Nan | Nan | 0.217 | 0.190 | 0.820 | 0.036 | 0.035 | 0.181 | 0.026 | 0.020 | 0.088 |
| M-ETHYLTOLUENE | 0.052 | 0.045 | 0.206 | 0.033 | 0.000 | 0.500 | 0.168 | 0.122 | 0.779 | 0.150 | 0.111 | 0.666 |
| M,P-XYLENE | 0.604 | 0.413 | 3.006 | 0.565 | 0.420 | 3.180 | 1.508 | 0.770 | 24.621 | 0.655 | 0.511 | 2.352 |
| METHYLCYCLOHEXANE | 0.074 | 0.056 | 0.344 | 0.003 | 0.000 | 0.460 | 0.187 | 0.085 | 2.100 | 0.189 | 0.064 | 4.341 |
| METHYLCYCLOPENTANE | 0.121 | 0.107 | 0.399 | 0.064 | 0.050 | 0.190 | 0.296 | 0.161 | 2.022 | 0.120 | 0.088 | 0.776 |
| N-BUTANE | 2.579 | 2.403 | 8.366 | 0.770 | 0.600 | 3.360 | 2.790 | 2.339 | 9.093 | 1.050 | 0.847 | 6.242 |

| | | | | | | | | | | | |
|---|---|---|---|---|---|---|---|---|---|---|---|
| **N-DECANE** | 0.021 | 0.018 | 0.093 | 0.074 | 0.070 | 0.270 | 0.108 | 0.071 | 0.544 | 0.086 | 0.068 | 0.206 |
| **N-HEPTANE** | 0.116 | 0.095 | 0.386 | 0.037 | 0.000 | 0.310 | 0.197 | 0.113 | 1.420 | 0.209 | 0.158 | 1.230 |
| **N-HEXANE** | 0.232 | 0.170 | 1.271 | 0.414 | 0.260 | 2.960 | 0.975 | 0.480 | 7.397 | 0.469 | 0.318 | 3.201 |
| **N-NONANE** | 0.033 | 0.026 | 0.187 | 0.057 | 0.050 | 0.270 | 0.079 | 0.048 | 0.434 | 0.469 | 0.361 | 1.607 |
| **N-OCTANE** | 0.046 | 0.037 | 0.191 | 0.100 | 0.080 | 0.440 | 0.107 | 0.072 | 0.720 | 0.091 | 0.078 | 0.244 |
| **N-PENTANE** | 0.877 | 0.762 | 2.383 | 0.508 | 0.480 | 1.280 | 0.751 | 0.626 | 3.083 | 0.936 | 0.657 | 7.593 |
| **N-PROPYLBENZENE** | 0.023 | 0.021 | 0.074 | 0.065 | 0.060 | 0.210 | 0.067 | 0.057 | 0.210 | 0.059 | 0.051 | 0.201 |
| **N-UNDECANE** | 0.033 | 0.030 | 0.136 | 0.011 | 0.000 | 0.190 | 0.094 | 0.073 | 0.396 | 0.133 | 0.115 | 0.291 |
| **O-ETHYLTOLUENE** | 0.024 | 0.021 | 0.086 | 0.058 | 0.050 | 0.200 | 0.078 | 0.057 | 0.322 | 0.067 | 0.055 | 0.263 |
| **O-XYLENE** | 0.175 | 0.126 | 0.933 | 0.256 | 0.200 | 1.270 | 1.058 | 0.633 | 8.043 | 0.327 | 0.256 | 1.176 |
| **P-DIETHYLBENZENE** | Nan | Nan | Nan | 0.000 | 0.000 | 0.000 | 0.071 | 0.050 | 0.563 | 0.054 | 0.042 | 0.151 |
| **P-ETHYLTOLUENE** | 0.030 | 0.025 | 0.127 | 0.043 | 0.040 | 0.160 | 0.107 | 0.076 | 0.478 | 0.086 | 0.070 | 0.329 |
| **PROPANE** | 3.651 | 3.456 | 11.666 | 2.355 | 2.130 | 9.360 | 4.801 | 3.754 | 20.957 | 1.221 | 1.121 | 2.860 |
| **PROPENE** | 0.581 | 0.496 | 2.472 | 0.897 | 0.870 | 3.430 | 0.568 | 0.358 | 3.387 | 0.785 | 0.717 | 2.336 |
| **STYRENE** | 0.040 | 0.026 | 0.383 | 0.202 | 0.170 | 1.270 | 0.180 | 0.079 | 2.078 | 0.119 | 0.084 | 0.493 |
| **TOLUENE** | 1.319 | 1.055 | 6.400 | 0.867 | 0.550 | 5.290 | 5.312 | 3.041 | 39.897 | 1.154 | 0.963 | 3.594 |
| **TRANS-2-BUTENE** | 0.110 | 0.092 | 0.470 | 0.003 | 0.000 | 0.260 | 0.201 | 0.186 | 0.395 | 0.175 | 0.125 | 1.851 |
| **TRANS-2-PENTENE** | 0.008 | 0.003 | 0.136 | 0.024 | 0.000 | 0.860 | 0.026 | 0.011 | 0.355 | 0.052 | 0.008 | 0.771 |

*In addition, I do not know what NO2 chemical conversion to NO means as stated on line 25 of page 3. This needs to be described and the probability of interference from PAN needs to be discussed. I would expect at least 5 ppbv of PAN in areas such as Beijing during the day. This could lead to a significant interference in NO2. Please also report in more detail on the VOC observations. I think at least averages of the top 10 or 5 VOC in terms of OH loss should be listed for each city. I like the graphs in the supplement but the scales on many of the graphs don't make sense. Often the parameter graphed only goes up to 10 or 20% of full scale making it impossible to see what is going on. I don't think keeping consistent axes between different cities is worth not being able to read the graph.*

**Answer:**

We changed the sentence to be "… after chemical conversion to NO." to be "in the form of NO by chemical conversion using Molybdenum convertor. This conversion method is known to be interference by $NO_z$ species, which could be converted to NO. Therefore, one should keep in mind that the $NO_2$ measurement presented in this study could be positive biased from the ambient $NO_2$ concentrations." Besides, we derived the PAN concentrations from our box model calculations, which are about 2 ppbv (Fig. S7).

[Figure]

**Figure S7. The mean diurnal profiles of modelled formaldehyde (HCHO), acetaldehyde (ACD) and peroxyacetyl nitrate (PAN) concentrations in four cities.**

We added a table about measured VOC in supplement (Table S3) and a table showing top 10 $k_{OH}$ contributing VOCs (Table 2). The mean diurnal profiles of top 10 VOCs are added in supplement (Fig. S6). A detail discussion on the measured VOCs is added. Please find the answers in the response to referee #1 who has the similar comments.

We changed the scale of the Fig. S1-S4 as suggested (see below).

[Figure]

**Figure S1 The time series of measured parameters (j(O$^1$D), Temperature, NO, NO$_2$, O$_3$, O$_x$, CO, AHC, isoprene) and modelled OH, HO$_2$, and RO$_2$ concentrations and OH reactivity in Beijing.**

[Figure]

**Figure S2 The time series of measured parameters (j(O$^1$D), Temperature, NO, NO$_2$, O$_3$, O$_x$, CO, AHC, isoprene) and modelled OH, HO$_2$, and RO$_2$ concentrations and OH reactivity in Shanghai.**

[Figure]

**Figure S3 The time series of measured parameters (j(O$^1$D), Temperature, NO, NO$_2$, O$_3$, O$_x$, CO, AHC, isoprene) and modelled OH, HO$_2$, and RO$_2$ concentrations and OH reactivity in Guangzhou.**

[Figure]

**Figure S4 The time series of measured parameters (j(O$^1$D), Temperature, NO, NO$_2$, O$_3$, O$_x$, CO, AHC, isoprene) and modelled OH, HO$_2$, and RO$_2$ concentrations and OH reactivity in Chongqing.**

*3) The reporting of OH reactivity could be made much simpler as well. Perhaps having a section in the results showing the VOC observations separately would be less confusing. You could then have a following section on the calculated OH reactivity. I really recommend limiting the discussion in these sections and focusing on the results. For example, the paragraph on line 1 page 5 stating that OH reactivity can be measured in 3 ways made me think for some time that this was a measured quantity in this work.*

**Answer:**

We removed the part of the description of OH reactivity measurement techniques. The description of VOC measurement and discussion are moved to section 4.1. Please see detail in the response to referee#1.

*4) I highly recommend being more explicit on what is derived from the model or parameterized. For example, I don't think formaldehyde or acetaldehyde are measured but are model predicted. If so this needs to be described and predicted levels compared to observations if available. This will certainly impact the radical budget as well as the production rate of PAN relative to HNO3. So I suggest a table of model parameters that are predicted, constrained, and parameterized. I also suggest that the model results be presented in an organized manner in the results section. There is a lot of discussion throughout section 3 that should probably be in section 4.*

**Answer:**

Similar to the comments (2), we prepared the table in supplement to state what are measured, modelled, and parameterized in this study (Table S2). We added a sentence in the new section 4.1 to discuss the modeled OVOC concentration "The OVOCs concentrations are simulated by the box model. The modelled HCHO concentrations were in the range of 3 to 8 ppbv (Fig. S7), which are consistent with the previous studies in these regions (Zhang et al., 2012;Song et al., 2018;Chen et al., 2016;Tang et al., 2009). The modelled acetaldehyde concentrations are in the range of 2 to 3 ppbv in Beijing, Shanghai, and Chongqing but on average 1 ppbv larger in Guangzhou because the larger contribution of aromatics VOCs which produce acetaldehyde from their OH degradation."

We restructured the manuscript by moving the VOC description to section 4.1 and moving the OH-HO$_2$-RO$_2$ budget analysis (originally section 3.3.2) to section 4.2.

*5) The very simple parameterization of HONO as being 2% of NO2 is somewhat troubling. I am surprised that it would be that simple especially as a function of the time of day. I think this assumption needs to be better justified and probably looked at to determine the sensitivity, i.e. some case studies with different assumptions are probably needed. This is also another reason to describe the NO2 measurement in more detail.*

**Answer:**

Referee #1 has similar concern on the uncertainty in HONO parameterization. We performed more sensitivity study to investigate the uncertainty and please find our answer in the response to Referee#1.

*6) I am not sure the ISOROPPIA modeling adds much to the paper especially as there are no measurements of ammonia or nitric acid. I certainly realize that if there is a large excess of ammonia that this will drive nitric acid into the aerosol. However, I am not sure the nitric formation rate vs. loss rate to aerosol versus dry and wet deposition can be suitably treated in this work to allow for quantitative predictions of ammonium nitrate aerosol. So I recommend removing from the paper and perhaps replacing with a simple discussion. This discussion could also mention that cutting down NOx may lead to enhanced ozone production but it will cut down on particulate nitrate as well.*

**Answer:**

We agree that the calculation may not be quantitative since some key parameters, e.g. ammonia and nitric acid concentrations were not measured during these campaigns. We reduce the content in the section Nitrate production potential. We also moved some of the contents and original Table 2 to supplement. Please find the changes in the main text following.

"Nitric acid is one of the major products generated by the radical system for high $NO_x$ conditions, which is an important precursor of particulate nitrate ($NO_3^-$). Recently, nitrate has become a significant portion in particles in Beijing, Shanghai, and Nanjing (YRD) during summertime in China (Li et al., 2018a). The gas phase nitric acid $HNO_3$ together with ammonium $NH_3$ form a gas-particle partitioning equilibrium with $NH_4NO_3$ (R3), which depends on the relative humidity, temperature, and the aerosol contents (Seinfeld and Pandis, 2016).

$$NH_3 + HNO_3 \leftrightarrow NH_4NO_3 \qquad\qquad (R3)$$

The nitric acid is mainly produced from the reaction between NO and OH which can be derived from the box model. The fate of nitric acid depends on the gas-particle partition, deposition and chemical reactions. In general, the time scale of partitioning is 1-2 orders smaller than those of deposition and chemical production (Morino et al., 2006;Neuman et al., 2003). Therefore, the photochemical produced nitric acid will deposit on to the aerosol if the ambient $NH_3$ is sufficient. The deposition rate is about 7 cm s$^{-1}$ (Seinfeld and Pandis, 2016), which results in a deposition timescale being 8 hours if the boundary layer height is 2 km (typical values for summertime). The ammonia concentrations are usually above 10 μg/m$^3$ in urban areas in China during summertime (Pan et al., 2018), which indicates ammonia-rich conditions and sufficient to neutralize nitric acid..

In this study, we use the aerosol thermodynamic model (ISORROPIA) to simulate the nitrate production and the model design is explained in supplement. It's worth noting that such model simulation cannot be quantitatively because some key parameters, e.g. ammonia and nitric acid concentrations were not measured during these campaigns. The discussion below should be considered as a qualitative estimation to show the important feature in determining the particle nitrate production. The modeled nitrate concentration and partitioning in Beijing are shown to illustrate the typical pattern of particulate nitrate formation (Fig. 11a). The total nitrate concentrations maximize in the late afternoon while the particulate nitrate shows a board peak at night, which is mainly driven by the stronger gas-to-particle partitioning due to higher RH. Since deliquesce relative humidity (DRH) of $NH_4NO_3$ is about 60% in all cases, the partitioning changed dramatically with the relative humidity above DRH (nighttime) and below DRH (daytime). One should note that the nitrate formation from $N_2O_5$ hydrolysis is not taken into account in this study, which could lead to negative bias in the nitrate production calculation.

To investigate the nitrate concentration dependence on the nitrate production rate and ambient ammonia concentrations, the averaged nitrate concentrations are plotted as a function of daily integrated nitric acid production rate and total ammonium ($NH_4^+{}_{(a)}+NH_{3(g)}$) concentrations. As shown in Fig. 11b, the isopleth diagrams are split into two parts by the dashed line to represent the nitrate- (upper left) and ammonium-sensitive (lower right) regimes. However, the threshold for nitrate- and the ammonium-sensitive regime is not distinct in the small chemical range. Actually, the nitrate concentrations are sensitive to both precursors. The daily integrated nitrate production rate and averaged total ammonium concentrations for each city are denoted by the circles (Fig. 11b). The circles are located above the ridgeline, which means nitrate concentrations are more sensitive to the change of nitric acid production rate. Therefore, this scenario study highlights that the further mitigation of summertime particulate nitrate pollution should aim at the reduction of photochemical nitric acid production. For example, the reduction in $NO_x$ emission could help to reduce the particulate nitrate pollution but may lead to enhancement in ozone pollution (see section 4.3). "

**Anonymous Referee #3**

*This paper presents results from a 0-D box model constrained by observations of radical sources and sinks in order to evaluate the oxidation capacity of several Chinese megacities, including Beijing, Shanghai, Guangzhou, and Chongqing. The models suggest that while there are similarities in the chemistry of each urban area, such as ozone production being VOC-limited in each, there are some distinct differences in predicted radical concentrations, rates of ozone production, and OH reactivity, which may help provide insights into specific control strategies for each area.*

*While the paper provides some interesting contrasts between the cities, it is unfortunately somewhat difficult to read due to issues related to both the amount of information and how it is presented as well as style and grammar. In particular, section 3.3.2 describing the radical budget analysis reads more like a stream of thought rather than an organized discussion.*

**Answer:**

The manuscript is restructured by moving the VOC description to section 4.1 and moving the OH-HO$_2$-RO$_2$ budget analysis (originally section 3.3.2) to section 4.2.

We have edited the manuscript substantially on the style and grammar and please find the modification in the revised manuscript.

*A major assumption in the paper is that the model can accurately reproduce concentrations of OH, HO2, and RO2 radicals in order to predict the oxidation capacity of each region. Unfortunately, there is no discussion of whether this is a reasonable assumption. As mentioned in the introduction, previous measurements of radical concentrations in urban areas often exhibit significant discrepancies with model predictions, suggesting that chemical models are unable to accurately reproduce the oxidation capacity of these areas (see for example Whalley et al. (2018), Griffith et al. (2016), in addition to references cited in the Introduction). As summarized in the Lu et al. (2018) review cited in the paper, ": : :current tropospheric chemical mechanisms cannot explain the OH radical concentrations in China, which strongly underestimated the OH concentrations and the local ozone production for the low and high NOx range, respectively." The authors should expand the discussion of these discrepancies and discuss in much more detail their potential impact on their model predictions and conclusions.*

**Answer:**

As shown in Rohrer, the OH is relative well captured by the model in moderate and high NOx regime. As the presented observation were conducted in megacities, which were mainly located in the high NOx regime. The model should be able to predict the OH concentrations well. The major question is the model underestimation for HO2 and RO2 concentrations and thus the local ozone production rate. We added

sentences in Line 6 Page 7 "As previous field campaign in China shown that the OH concentrations could be underestimated in the low $NO_x$ conditions (Tan et al., 2018b;Tan et al., 2017;Fuchs et al., 2017;Rohrer et al., 2014;Lu et al., 2013;Lu et al., 2012;Hofzumahaus et al., 2009). In this study, the $NO_x$ concentration are in moderate and high range, where the model is capable to reproduce the OH concentrations relatively good (Rohrer et al., 2014). For the high $NO_x$ regime, the prominent feature is the underestimation of $HO_2$ and $RO_2$ concentrations (Tan et al., 2018a;Tan et al., 2017;Tan et al., 2018d). This is also found in other urban site outside China (Griffith et al., 2016;Whalley et al., 2018;Kanaya et al., 2007;Dusanter et al., 2009;Shirley et al., 2006;Brune et al., 2016;Ren et al., 2013), indicating a common defect in current chemical mechanisms. Such model defect will lead to underestimation of local ozone production. The explanation of model underestimation is out of the scope of this study but the possible impact will be discussed in section 4.3."

We also added the discussion on potential underestimation of $P(O_3)$ in Line 22 Page 12 "As mentioned in section 3.3, the current model could have defects for high $NO_x$ conditions, which underestimated the peroxy radical concentrations and thus local ozone production China (Tan et al., 2017;Griffith et al., 2016;Whalley et al., 2018;Kanaya et al., 2007;Dusanter et al., 2009;Shirley et al., 2006;Brune et al., 2016;Ren et al., 2013). However, the quantitative estimation is not possible due to the absence of in-situ radical measurements. To our knowledge, no field campaigns have been conducted to perform in-situ radical measurements in city center area in China. However, a field campaign in downwind area of Beijing (YUFA) found local ozone production rate was underestimated due to the underestimation of $HO_2$ concentrations (Lu et al., 2010). Another field campaign in a rural site in NCP also found model underestimation of $P(O_3)$ by 20 ppbv per day compared a daily integrated ozone production of 110 ppbv derived from the measured $HO_2$ and $RO_2$ (Tan et al., 2017). Therefore, the ozone production rate presented derived from model calculations in this study should be considered as a lower limit. Nevertheless, the underestimation of peroxy radical concentration will not affect the $O_3$-$NO_x$-VOC sensitivity diagnosis (Tan et al., 2018d)."

Additional comments:

*1) The paper would benefit from a more detailed description of what was measured and how they were measured, perhaps with a table in the supplement. In particular, the specific VOCs that were measured should be described in more detail.*

**Answer:**

We added a table to describe the instrumentation in the supplement (Table S1). We also prepared the table in supplement to state what are measured, modelled, and parameterized in this study (Table S2). The measured VOCs and their concentrations are presented in table S3.

*2) Instead of just showing the total AHC (or preferably AVOC as indicated elsewhere in the manuscript), it would be more informative to illustrate the diurnal mixing ratios of some important individual VOCs that demonstrate the similarities and differences in the areas as described in the manuscript.*

**Answer:**

We changed all AHC to be AVOC in the revised manuscript.

We added a table about measured VOC in supplement (Table S3) and a table showing top 10 $k_{OH}$ contributing VOCs (Table 2). The mean diurnal profiles of top 10 VOCs are added in supplement (Fig. S6). A detail discussion on the measured VOCs is added. Please find the answers in the response to referee #1 who has the similar comments.

*3) In addition, it should be clarified which VOCs and/or OVOCs were measured and which were modeled as part of the radical budget. For example, were HCHO and other carbonyls measured or was their contribution to radical production based on modeled concentrations?*

**Answer:**

The VOC are measured and OVOCs are modelled. Therefore, the alkene ozonolysis is observation constrained. We make this point clear by adding a sentence in the radical budget analysis section ""

We added the discussion of modelled OVOCs results in the end of section 4.1 VOC compositions and ozone production efficiency in the revised manuscript. The modelled OVOCs concentrations are comparable to previous studies for these regions, indicating the model is capable to reproduce the OVOCs formation. The added content is "The OVOCs concentrations are simulated by the box model. The modelled HCHO concentrations were in the range of 3 to 8 ppbv (Fig. S7), which are consistent with the previous studies in these regions (Zhang et al., 2012;Song et al., 2018;Chen et al., 2016;Tang et al., 2009). The modelled acetaldehyde concentrations are in the range of 2 to 3 ppbv in Beijing, Shanghai, and Chongqing but 1 ppbv larger in Guangzhou because the larger contribution of aromatics VOCs which produce acetaldehyde from their OH degradation."

[revised manuscript text omitted]

**1. Introduction**

Air pollution is the one of the major threats to human health in cities (Kan et al., 2012). In China, the rapid economic development accompanied by degradation of air quality for the last decades in the eastern areas (Chan and Yao, 2008). More than 300 million people live in the North China Plain (NCP), Yangtze River Delta (YRD) and Pearl River Delta (PRD) regions in
5 eastern China. Among all, Beijing, Shanghai, and Guangzhou are the metropolitan cities in these regions and suffering from severe air pollution. The Chengdu-Chongqing city group (population 90 million) locates in Sichuan Basin (SCB), southwest of China, representing the developing city clusters. Chongqing is the biggest city in the southwest of China, which suffers from severe air pollution as well. 
[revised manuscript text omitted]
 be interference by $NO_z$ species, which could be converted to NO. Therefore, one should keep in mind that the $NO_2$ measurement presented in this study could be positive biased from the ambient $NO_2$ concentrations. CO was measured by the infrared absorption technique using Thermo instrument (Model 20). The performance of different instruments is summarized in Table S1. Speciated VOCs measurement was performed by the GC-MS/FID. VOC measurements (including 55 organic species) were performed by

10 commercial an instrumentation using gas chromatograph (GC) equipped with a mass spectrometer (MS) and a flame ionization detector (FID). In principle, the air sample was drawn into two parallel channels for enrichment by cooling before analysis (Wang et al., 2014). The VOCs measurements include $C_2$–$C_{11}$ alkanes, $C_2$–$C_6$ alkenes, and $C_6$–$C_{10}$ Aromatics (Table S2). The photolysis frequencies were measured by spectrum radiometer. Meteorological parameters were measured simultaneously, e.g. ambient temperature, pressure, and relative humidity.

15 **2.3 The model**

A box model based on the Regional Atmospheric Chemical Mechanism version 2 (Goliff et al., 2013) is used to simulate the concentrations of the  OH, $HO_2$ and $RO_2$ radicals concentrations and other unmeasured secondary species concentrations. The newly proposed isoprene mechanisms are also incorporated (Peeters et al., 2014;Fuchs et al., 2013). The model was constrained to the observation of photolysis frequencies, long-lived trace gases (NO, $NO_2$, $O_3$, CO, $C_2$−$C_{12}$ VOCs), and meteorological

20 parameters. Since nitrous acid (HONO) was not measured in these campaigns, it was fixed to 2% of the observed $NO_2$ concentrations because good correlation was found between HONO and $NO_2$ in different field studies with a constant ratio being 0.02 (Elshorbany et al., 2012). The uncertainty of such parameterization is discussed in section 4.2. The measured, modelled and parameterized parameters are summarized in Table S3. The uncertainty of the model calculations depends on the model constraints and the reaction rate constants. Taking into account the uncertainties of both measurements and

25 kinetic rate constants, the model calculations is approximately 40% (Tan et al., 2017).

**3. Results**

**3.1 Overview of measurements**

The mean diurnal profiles of measured ambient temperature, $j(O^1D)$, and CO, $O_3$ ($O_x = O_3 + NO_2$), $NO_x$(=NO+$NO_2$), and AVOC concentrations are shown in Fig. 2 (the time series are shown in Fig. S1-S4). The ambient temperature is relatively

30 similar in Beijing, Shanghai, and Chongqing, but lower in Guangzhou because the campaign was conducted in a later time of a year. Similarly, the photolysis frequencies are smaller in Guangzhou. However, $j(O^1D)$ is highest in Beijing and are comparable in Shanghai and Chongqing. However, diurnal maximum $O_3$ concentrations are highest in Shanghai (80 ppbv) followed by Beijing (72 ppbv), Guangzhou (65 ppbv), and Chongqing (56 ppbv). The diurnal peak of $O_3$ appears at 15:00~16:00 LT in Beijing, Guangzhou, and Chongqing. In Shanghai, the peak of $O_3$ shows up at 13:00 LT due to the fast increase in the morning. During the

35 measurement period, the observed ozone concentrations exceed the Chinese National Air Quality Standard Grade II (99.3 ppbv) in Beijing, Shanghai, and Guangzhou (Table 1).

When a measurement site is close to $NO_x$ emission sources, part of the $O_3$ is titrated to $NO_2$ by fresh emitted NO. Although $O_3$ is regenerated in a few minutes to half hour after the photolysis of $NO_2$, $O_3$ is stored temporally in the form of $NO_2$. Therefore, $O_x$, the sum of $O_3$ and $NO_2$, is a better metric to describe ozone pollution in the urban area. $O_x$ concentrations are also shown with broken lines in Fig. 2. In this case, the $O_x$ mean diurnal profiles in Beijing, Shanghai and Guangzhou show maximum values of

5 about 90 ppbv (1-hour resolution), indicating the ozone pollution are comparable in these cities during the measurement period. In Chongqing, the maximum of the diurnal average is 66 ppbv. The ozone pollution is serious in autumn in Guangzhou due to the unique synoptic system, including the surface high-pressure system, hurricane movement and the sea-land breeze (Fan et al., 2008). In Shanghai, the synoptic weather is crucial to pollution accumulation, and the ozone concentrations are reduced in August and September due to the cleaning effect by the summer Monsoon (Dufour et al., 2010;Geng et al., 2015).

10 Given the relatively short periods for these campaigns, one concern is about the representativeness of measurements. We compared the observation from these intensive campaigns to the routine measurement obtained in the environmental monitor stations operated by the Chinese environmental protection agency (Fig. S5). We found that the mean diurnal profiles of $O_3$ and $O_x$ obtained in all sites are comparable to the highest monthly averaged diurnal profiles for the same city (bias < 20%). The relatively small $O_3$ and $O_x$ concentrations observed in Chongqing compared to other cities (Fig. 2) is consistent with the environmental monitor stations

15 observation (Fig. S5). Therefore, it suggests that the ozone pollution is less severe in Chongqing compared to the megacities in eastern China.

[revised manuscript text omitted]

15 3.3 The top 10 OH reactivity contributing VOCs are summarized in Table 2. The order of importance is sorted by the averaged OH reactivity for four cities. Among all, propene are the most important OH reactivity contributor, which contributed about 0.4~0.6 s$^{-1}$ (Table 2). The small VOCs (propene, ethane, ethene) are relatively important with respect to OH reactivity. 9 out of the top 10 VOCs are alkenes and aromatics (except ethane). In Guangzhou, the xylene (m,p-, and o-) and toluene are also important OH reactants, consistent with the inventory study (Zheng et al., 2009). The diurnal profiles are shown in Fig. S6. The observed

20 anthropogenic VOC concentrations show typical diurnal profile that increase during night and decrease during afternoon. One exception is Shanghai site, the mean diurnal profiles of propene and 1,2,4-trimethybenzene are flat, while that of styrene shows peak around noontime, indicating unique VOC emission feature in that site.

The OVOCs concentrations are simulated by the box model. The modelled HCHO concentrations are in the range of 3 to 8 ppbv

25 (Fig. S7), which are consistent with the previous studies in these regions (Zhang et al., 2012;Song et al., 2018;Chen et al., 2016;Tang et al., 2009). The modelled acetaldehyde concentrations are in the range of 2 to 3 ppbv in Beijing, Shanghai, and Chongqing but 1 ppbv larger in Guangzhou because the larger contribution of aromatics VOCs which produce acetaldehyde from their OH degradation.

**4.2 OH-HO$_2$-RO$_2$ Radical budget analysis**

30 All radical reactions are classified into four groups (initiation, termination, propagation, and thermos-equilibrium with reservoir species). The reaction turnover rate illustrates the important processes in the RO$_x$ radical reactions framework. The initiation and termination rate are shown in Fig. 7. The following radical budget analysis will focus on the daytime conditions (06:00—18:00) if no additional clarification.

The dominant radical sources are photolysis reactions, including HONO, O$_3$, HCHO and other carbonyl compounds. The photolysis

35 of HONO and O$_3$ (producing O$^1$D and followed by H$_2$O reaction) produce OH radicals, which contributes 33-45% to the total primary source, P(RO$_x$). among all campaigns. The HCHO photolysis produces HO$_2$ (14-33% of P(RO$_x$)) while the other carbonyl compounds photolyze to generate RO$_2$ radicals (3-6% of P(RO$_x$)). Therefore, photolysis reactions dominate the radical primary sources during daytime (58-86%). In contrast, alkenes ozonolysis is the dominant radical source during nighttime and the yields of OH, HO$_2$, and RO$_2$ radicals depend on individual alkenes. The maximum of P(RO$_x$) mean diurnal profile is largest in Beijing (5

ppbv/h), followed by Shanghai (4.6 ppbv/h) and Chongqing (4.3 ppbv/h). The daytime averaged $P(RO_x)$ is smaller in Shanghai due to the narrower peak of photolysis frequencies (Fig. 2) and shorter photolysis reaction time. The primary radical source is smallest in Guangzhou (3.2 ppbv/h) due to the later observation period in a year. However, the alkene ozonolysis contributed significantly to the radical sources in Guangzhou (43% of the total primary source for daytime conditions), which could attribute

5   to higher abundance of alkenes due to special emission inventory (see section 3.2).

The ozonolysis reactions mainly contributed by trans-2-butene in Beijing (55%), Guangzhou (42%), and Chongqing (39%), whose concentrations are in the range between 0.1 to 0.3 ppbv (Fig. S6). Although trans-2-butene is only the 8th important VOCs with respect to OH reaction (Table 2), it become the most important $O_3$ reactants producing $RO_x$ radicals due to its fast reaction rate with $O_3$ ($1.9\times10^{-16}$ $cm^{-3}s^{-1}$ compared to $1.0\times10^{-17}$ $cm^{-3}s^{-1}$ of propene, rate constant derived from MCM3.3.1

10  (http://mcm.leeds.ac.uk/MCMv3.3.1/home.htt)). In Shanghai, propene becomes the most important alkene with respect to $O_3$ reaction, which accounts for about 42% of the total alkene ozonolysis reactions. Actually, the relatively high contribution from alkene ozonolysis to the $RO_x$ radical primary sources could be one of the important characteristics for $RO_x$ radical primary sources in Chinese megacity. The importance of alkene ozonolysis was also found in Santiago, Chile (Elshorbany et al., 2009) and Essex, UK (Emmerson et al., 2007), where alkene ozonolysis contributed about 20% to the total radical primary production.

15  Radical termination can be divided into two groups, the nitrogen-containing compounds, including HONO, $HNO_3$, $RONO_2$, and PAN-type species ($L_N$). The other pathway leads to peroxides formation result from the combination of two peroxy radicals ($L_H$). The ratio between $L_N$ and $L_H$ depends on the $NO_x$ concentrations. In urban environments, the limiting factor for radical propagation is the abundance of VOCs. In our case, the radical termination is dominated by $L_N$ (>70%). Among all, the nitric acid formation was the major contributor to the radical termination in all cities (>50%). In Chongqing, the peroxide formation path contributes

20  26% to the radical termination, especially the ratio increase to 32% during the afternoon due to the higher VOC/$NO_x$ ratio. Net PAN-type species formation as a radical loss becomes relatively important in Guangzhou (about 20%) due to the lower temperature (Fig. 2). It is reported that on average 25% of the radical can be lost via forming PAN-type species in Beijing during winter (Tan et al., 2018c). Besides, PAN-type species formation becomes important in the urban area, e.g. it contributes 30% to the total radical loss in London downtown area (Whalley et al., 2018).

25  The comparison of the four cities is clearly shown in Fig. 8. The HONO photolysis is the dominant OH source in all cities except in Shanghai. The $O_3$ photolysis is more important than HONO photolysis in Shanghai, contributing 55% to the total OH primary sources and 23% of the total radical sources. In all cities, the primary production of $HO_2$ is comparable to that of OH, which is mainly contributed by the HCHO photolysis and alkene ozonolysis. These results are consistent with the model calculation performed in Beijing (Yang et al., 2018) and Hong Kong (Xue et al., 2016). The importance of HONO and HCHO photolysis to

30  radical primary production is also found in suburban and rural environments (Lu et al., 2012;Lu et al., 2013;Tan et al., 2017). In the base model scenario, HONO is scaled to $NO_2$ measurements and the uncertainty of this assumption is further discussed following. In the base model, the HONO concentrations are scaled to the observed $NO_x$ concentration using a scaling factor 0.02 (Elshorbany et al., 2012). In this study, we use this scaling factor between HONO and $NO_x$ to simplify the discussion of unknown HONO sources. In the original RACM2 model, only homogenous source is included, i.e. OH+NO→HONO, which is not sufficient

35  to support the high daytime HONO concentrations and, as a result, leads to a strong underestimation of OH concentrations (Su et al., 2011;Yang et al., 2014;Tong et al., 2016;Ye et al., 2016;Li et al., 2012;Li et al., 2014b). Although the HONO to $NO_2$ ratio is relative robust and constant as reported in other field campaigns (Elshorbany et al., 2012), such simple parameterization could increase the uncertainty of our model calculation. To further investigate the uncertainty from this simple parameterization, the scaling factor is varied from 0.015 to 0.03. The modelled OH concentrations change by less than 10 % if the HONO scaling factors

40  change by 50% (Fig. S8). Besides, the modelled $HO_2$ and $RO_2$ concentrations are relatively stable with different HONO scaling

factors. The different scaling factors also have impact on the model generated species, e.g. HCHO (Table S4). In fact, the higher HONO concentrations lead to more active photochemical reactions and more HCHO production. The higher HCHO concentrations could further enhance the photochemistry by more radical photolytic sources in return. Therefore, the higher (lower) modelled radical concentrations due to increase (reduce) the HONO scaling factors are also affected by the corresponding change in modelled

5    HCHO concentrations. This demonstrates the nonlinearity of the photochemical system. Nevertheless, the parameterized HONO concentrations are in the range of 0.3 to 0.6 ppbv during daytime (Table S4), which are consistent with previous in-situ measurements in urban areas (Lu et al., 2013;Li et al., 2010;Ren et al., 2003;Kanaya et al., 2007). To evaluate the impact of missing HONO source on the radical chemistry, we switched off the scaling between HONO and $NO_x$ in a sensitivity test. (Fig. S8). Therefore, the results show that OH concentrations reduce by about 20% if the only homogenous source is considered. The modeled

10   $HO_2$ and $RO_2$ concentrations are also reduced  15-20 % (Table S4).

15   ~~types species tends to result in a radical loss (0.1~0.3 ppbv/h). The equilibrium between $HO_2$ and $HNO_4$ is fast with null effect on~~

The $RO_2$ primary source strength is in the range of 0.2 to 0.3 ppbv/h, which is mainly contributed by alkene ozonolysis and OVOC photolysis (Fig. 8). In this study, the OVOC photolysis mainly includes carbonyl-containing compounds (e.g. acetaldehyde, aldehydes with carbon numbers larger than 3), which are generated by the box model. The modelled acetaldehyde concentrations

20   are in the range of 2 to 4 ppbv (Fig. S7), consistent to the observations in Beijing (Chen et al., 2016) and Hong Kong (Lyu et al., 2016). The photolysis rate of carbonyl-containing compounds (except HCHO) is about one third to a quarter of the HCHO photolysis rate. In comparison, this ratio could be close to or even higher than 1 in other urban studies (Ren et al., 2013;Volkamer et al., 2010;Emmerson et al., 2007;Michoud et al., 2012;Whalley et al., 2018;Xue et al., 2016). In contrast, the relatively small contribution from other carbonyl-containing compounds photolysis than HCHO photolysis were reported in an urban and suburban

25   site in Hong Kong (Lyu et al., 2016), where the acetaldehyde concentrations were about 1 to 2 ppbv, comparable to our model simulation. Such large discrepancy in the role of other OVOC photolysis to the radical production highlights the importance to measure these radical precursors in the 
[revised manuscript text omitted]
 defects for high $NO_x$ conditions, which underestimated the peroxy radical concentrations and thus local ozone production China (Tan et al., 2017;Tan et al., 2018d;Griffith et al., 2016;Whalley et al., 2018;Kanaya et al., 2007;Dusanter et al., 2009;Shirley et al., 2006;Brune et al., 2016;Ren et al., 2013). However, the quantitative estimation is not possible due to the absence of in-situ radical measurements. To our knowledge, no field campaigns 10 have been conducted to perform in-situ radical measurements in city center area in China. However, a field campaign in downwind area of Beijing (YUFA) found local ozone production rate was underestimated due to the underestimation of $HO_2$ concentrations (Lu et al., 2010). Another field campaign in a rural site in NCP also found 
[revised manuscript text omitted]
 and the model design is explained in supplement. It's worth noting that such model simulation cannot be quantitatively because some key parameters, e.g. ammonia and nitric acid concentrations were not measured during these campaigns. The discussion below should be considered as
15 a qualitative estimation to show the important feature in determining the particle nitrate production. The modeled nitrate concentration and partitioning in Beijing are shown to illustrate the typical pattern of particulate nitrate formation (Fig. 11a). The total nitrate concentrations maximize in the late afternoon while the particulate nitrate shows a board peak at night, which is mainly driven by the stronger gas-to-particle partitioning due to higher RH. Since deliquesce relative humidity (DRH) of NH₄NO₃ is about 60% in all cases, the partitioning changed dramatically with the relative humidity above DRH (nighttime) and
20 below DRH (daytime). One should note that the nitrate formation from N₂O₅ hydrolysis is not taken into account in this study, which could lead to 
[revised manuscript text omitted]

| VOCs | Beijing | Shanghai | Guangzhou | Chongqing |
|------|---------|----------|-----------|-----------|
| PROPENE | 0.40 | 0.61 | 0.40 | 0.52 |
| ISOPRENE | 0.64 | 0.00 | 0.31 | 0.92 |
| ETHANE | 0.58 | 0.29 | 0.24 | 0.59 |
| ETHENE | 0.41 | 0.17 | 0.29 | 0.73 |
| M,P-XYLENE | 0.29 | 0.27 | 0.74 | 0.31 |
| TOLUENE | 0.18 | 0.12 | 0.73 | 0.15 |
| STYRENE | 0.06 | 0.28 | 0.26 | 0.16 |
| TRANS-2-BUTENE | 0.17 | 0.00 | 0.31 | 0.26 |
| O-XYLENE | 0.06 | 0.08 | 0.35 | 0.10 |
| 1,2,4-TRIMETHYLBENZENE | 0.08 | 0.13 | 0.16 | 0.17 |

[Figure]

**Figure 1. The location of the four measurement sites in Chinese megacities.**

[Figure]

[Figure]

**Figure 2. Mean diurnal variation of measured temperature, CO, O₃, j(O¹D), NOₓ and Anthropogenic Volatile Organic Compounds (AVOC) in four field studies. Oₓ is denoted in the same panel as O₃ with dashed lines.**

[Figure]

**Figure 3. Mean diurnal profile of contributions from all measured species for OH reactivity in Beijing, Shanghai, Guangzhou, and Chongqing. The filled areas represent different atmospheric constituents.** The model denotes the sum of model generated species such as formaldehyde, acetaldehyde.

[Figure]

**Figure 4. Contributions of different atmospheric constituents to OH reactivity in Beijing, Shanghai, Guangzhou, and Chongqing.** The OVOCs contributions are simulated by a box model.

[Figure]

**Figure 5. The group compositions (mixing ratios) in percentages for VOCs as well as their shares in OFPs and OH reactivity for four cities.**

[Figure]

5    **Figure 6.**

[Figure]

**Figure 5.** Mean diurnal profiles of modeled OH, HO₂, RO₂ concentrations in four measurement sites. **The vertical bars denote the daily variability of model calculated radical concentrations.**

[Figure]

Figure 6. The group compositions (mixing ratios) in percentages for primary VOCs as well as their shares in OFPs and OH reactivity for four cities

[Figure]

**Figure 7. Hourly averaged primary sources and sinks of RO$_x$ radicals derived from model calculations in four measurement sites.**

[Figure]

[Figure]

**Figure 8. Comparison of OH-HO₂-RO₂ radical budget in four cities for daytime conditions (06:00-18:00). The numbers are sorted in the order of Beijing, Shanghai, Guangzhou, and Chongqing from left to right. Blue boxes denote radical primary sources, black boxes denote radical termination, red boxes denote radical propagation, and yellow boxes denote equilibrium between radicals and reservoir species.**

[Figure]

[Figure]

**Figure 9. In situ ozone budget analysis at four cities. The upper panel denotes the local ozone production rate P(O₃) derived from model calculation. The middle panel denotes the derivatives of observed Oₓ concentrations d(Oₓ)/d(t). The bottom panel denotes the difference between P(O₃) and d(Oₓ)/d(t), which indicate the role of local chemical production on transportation (see text).**

[Figure]

[Figure]

**Figure 10. The RIR analysis for NOx, AVOC, CO and NVOC (natural volatile organic compounds, in this study only isoprene is considered) at four sites.**

[Figure]

Figure 11. Modelled nitrate production from gas-phase oxidation. (a) The mean diurnal profile of modelled total nitrate concentration and its gas-particle partitioning. (b) Functional dependence of particulate nitrate concentrations on daily integrated nitrate production rate and total ammonium concentrations.

[Figure]

Figure 12 Inter-comparison of atmospheric oxidation rate for four megacities.